# The influenza virus PB2 protein evades antiviral innate immunity by inhibiting JAK1/STAT signalling

Hui Yang [1], Yurui Dong[1], Ying Bian[1], Nuo Xu[1], Yuwei Wu[1], Fan Yang[1], Yinping Du[1], Tao Qin[1,2,3,4], Sujuan Chen[1,2,3,4] ✉, Daxin Peng [1,2,3,4] ✉ & Xiufan Liu[1,2,3,4]

Influenza A virus (IAV) polymerase protein PB2 has been shown to partially inhibit the host immune response by blocking the induction of interferons (IFNs). However, the IAV PB2 protein that regulates the downstream signaling pathway of IFNs is not well characterized. Here, we report that IAV PB2 protein reduces cellular sensitivity to IFNs, suppressing the activation of STAT1/STAT2 and ISGs. Furthermore, IAV PB2 protein targets mammalian JAK1 at lysine 859 and 860 for ubiquitination and degradation. Notably, the H5 subtype of highly pathogenic avian influenza virus with I283M/K526R mutations on PB2 increases the ability to degrade mammalian JAK1 and exhibits higher replicate efficiency in mammalian (but not avian) cells and mouse lung tissues, and causes greater mortality in infected mice. Altogether, these data describe a negative regulatory mechanism involving PB2-JAK1 and provide insights into an evasion strategy from host antiviral immunity employed by IAV.

Influenza A viruses (IAVs) are common contagious pathogens involved in global pandemics and seasonal epidemics, persistently threatening animal and human health[1]. In the past century, IAVs (H1N1, H2N2, and H3N2), which all originated in whole or partly from non-human reservoirs, have caused four influenza pandemics and constant infection in the human population. Due to the host species barrier, it is difficult for the avian influenza virus (AIV) to spread from birds to humans directly. However, the emergence of H7N9 and H5Nx (i.e., H5N1, H5N6, and H5N8) AIV subtypes have caused hundreds of severe or fatal human cases and are of particular concern[2–5]. The H7N9 viruses were first detected in live poultry markets in China, and there have been over 1500 human infections caused by the H7N9 virus, with a mortality rate reaching 40%[6]. Human cases of the highly pathogenic H5N1 AIVs continue throughout parts of Asia, Africa, and Europe, causing several hundred human cases and deaths since 2003[7,8]. To date, 78 laboratory-confirmed cases of human infection with H5N6 viruses have been reported, including 33 deaths[9]. H5N8 AIV has emerged in Asia and Europe since 2014, causing massive deaths in poultry and wild birds[5,10]. The first case of humans infected with H5N8 viruses was reported in Russia in 2021[11], although fatal human cases of the H5N8 subtype have not been documented as yet.

Host adaptation requires the viruses to overcome restriction barriers and replicate efficiently in a new host. To overcome host barriers, viruses must gain specificity for binding to cell receptors, adapt to host machinery for genome replication, transcription, and protein synthesis in cells, and evade restriction by elements of the host innate immune system during virus infection. AIVs obtain a series of mutations in their genomes to adapt to mammalian hosts before efficiently replicating and transmitting among humans. The PB2 protein of IAV is critical for viral replication and a key determinant of the host range[12]. The PB2 E627K substitution is a dominant adaptation marker in most human-adapted IAVs, facilitating replication in mammalian cells. Furthermore, the 590S/591R motif was found to complement the function of 627K in the PB2 of the 2009 H1N1 virus to allow replication in humans[13]. In addition, the 627K marker and, to a lesser extent, the 701N substitution have been described as critical for replicating H5N1

[1]College of Veterinary Medicine, Yangzhou University, 225009 Yangzhou, Jiangsu, China. [2]Jiangsu Co-Innovation Center for the Prevention and Control of Important Animal Infectious Disease and Zoonoses, 225009 Yangzhou, Jiangsu, China. [3]Joint International Research Laboratory of Agriculture and Agri-Product Safety, the Ministry of Education of China, Yangzhou University, 225009 Yangzhou, Jiangsu, China. [4]Jiangsu Research Centre of Engineering and Technology for Prevention and Control of Poultry Disease, 225009 Yangzhou, Jiangsu, China. ✉e-mail: chensj@yzu.edu.cn; pengdx@yzu.edu.cn

and H7N7 AIVs in mammalian hosts[14,15]. The host protein, ANP32A, has recently been found to play a role in restricting virus replication in mammalian cells through differential regulation of the activity of viral polymerases carrying PB2-627K (human) or PB2-627E (avian) signature[16,17]. 526R is another marker of mammalian adaption by AIVs, possibly functioning through interaction between viral PB2 and nuclear export protein (NEP) during virus replication[18]. We previously found that the synergistic effect of amino acid residues 283M and 526R in PB2 enhances the virulence of HPAI H5 viruses in mice by affecting the activity of the polymerase[19]. However, the effect of PB2 host-adaptive mutations is not limited to enhancing viral polymerase activity.

The innate immune system is the first line of host defense against viral infection, and interferons (IFNs) are recognized as an essential component of the innate immune response[20,21]. Janus kinase/signal transducer and activator of transcription (JAK/STAT) signal transduction pathway are important in controlling immune responses, which mediates the functions of IFNs and other cytokines[22]. The IFNs are secreted and bind to the cognate IFN receptor (IFNAR) to activate the JAK/STAT signaling. JAK1 and tyrosine kinase 2 (Tyk2) phosphorylate STAT1/STAT2, forming the dimmer of STAT1/STAT2 and translocation into the nucleus and binding to a DNA element, thereby specifically triggering the transcription of a group of IFN-stimulated genes (ISGs). These antiviral genes play an important role in establishing the antiviral response in host cells[23].

It is known that the binding of virus-induced cytokines to their receptors activates the JAK/STAT signaling. Although the JAK/STAT pathway is well-documented to be mainly activated through interferons for antiviral host defenses, many viruses have evolved mechanisms to combat antiviral defenses and establish an efficient infection. SARS-CoV-2 targets JAK1, Tyk2, and IFNAR subunit 1(IFNAR1) resulting in cellular desensitization to type I IFN[24]. The foot-and-mouth disease virus (FMDV) degrades JAK1 via the lysosomal pathway, inhibiting the IFNγ signaling transduction pathway[25]. Zika virus (ZIKV) targets JAK1 for proteasomal degradation, suppressing JAK/STAT signaling and impairing interferon-mediated antiviral response[26]. Human cytomegalovirus (HCMV) inhibits the IFNα signal transduction pathway by inducing proteasome-dependent degradation of JAK1[27]. Our previous work suggests that IAV infection induces SOCS1 expression to promote JAK1 degradation, inhibiting host innate immune responses[28].

Ubiquitination is a post-translational modification process critical for cellular regulating processes such as protein degradation and signaling transducing pathways. Viruses, including IAV, have hijacked the ubiquitin-proteasome system (UPS) to evade innate immune responses and promote proper infection[29–31]. IAV NS1 protein interacts with TRIM25 and then regulates host immune gene expression[32,33]. Hemagglutinin of IAV leads to the ubiquitination and degradation of IFNAR1[34]. This study describes how the IAV PB2 protein negatively regulates IFNs response. Intriguingly, this involves direct ubiquitination and degradation of the JAK1, a mechanism not yet described for any other influenza viral protein.

## Results
### IAV PB2 inhibits IFN-mediated signaling
To elucidate whether IAV PB2 protein may regulate IFN-mediated induction of antiviral genes, we chose two representative strains A/Puerto Rico/8/34 (H1N1, PR8) and A/goose/Eastern China/CZ/2013 (H5N8, CZ), to construct 3xFlag-tagged expression plasmids encoding their PB2 protein. We then explored their ability to regulate the activation of the IFNβ, ISRE, and NF-κB promoter triggered by Sendai virus (SeV) (Fig. 1a) infection or transfected cytoplasmic poly(I:C), a synthetic analog of viral dsRNA (Fig. 1b). The results showed that PB2 inhibited SeV, or poly(I:C)-induced activation of the IFNβ, ISRE, or NF-κB promoter (Fig. 1a, b). Overexpression of PB2 also inhibited the activation of ISRE and STAT1 promoter triggered by IFNβ (Fig. 1c) or

IFNα (Fig. 1d). Quantitative PCR (qPCR) analysis indicated that overexpression of PB2 significantly inhibited IFNs-triggered transcription of the interferon-induced protein with tetratricopeptide repeats 1 (IFIT1) and IFN-stimulated gene 15 (ISG15) in human A549 cells (Fig. 1e, f). Since PB2 inhibited the transcription of virus-triggered host genes, we next investigated whether PB2 plays a role in the cellular antiviral response. As shown in Fig. 1g, ultraviolet radiation-inactivated supernatants from PB2-transfected HEK293T cells which were infected with SeV could not completely inhibit the replication of GFP-expressing vesicular stomatitis virus (VSV-GFP) as monitored by the GFP signal, which suggested that PB2 dampened the secretion of antiviral factors induced by SeV (Fig. 1g and Supplementary Fig. 1). Further, IFNs-induced phosphorylation of STAT1 and STAT2 were markedly inhibited by PB2 (Fig. 1h, i). Altogether, these data suggest that PB2 protein is a generally negative regulator of the IFNs-triggered antiviral response.

### IAV PB2 promotes the degradation of JAK1 through the proteasome machinery
We next investigated how PB2 inhibits the IFN-induced antiviral response. Indeed, The PB2 proteins from PR8 or CZ could down-regulate endogenous JAK1 protein levels, but not IFNAR1 and STAT1 in HEK293T cells (Fig. 2a and Supplementary Fig. 2a). Additionally, when HEK293T cells were transfected with JAK1-His or Flag-STAT1 along with increasing amounts of PB2 from PR8 or CZ, we observed dose-dependent reductions in the levels of JAK1-His (Fig. 2b and Supplementary Fig. 2b), but not STAT1 (Fig. 2c and Supplementary Fig. 2c). In contrast, PB2 expression did not alter the mRNA level of JAK1 (Fig. 2d). Following PR8 or CZ infection, the expression level of JAK1 was decreased in A549 cells (Fig. 2e and Supplementary Fig. 2d). Further, we employed short hairpin RNA (shRNA)-mediated knockdown of PB2 expression, and we generated two shPB2 stable HEK293T cell lines (Supplementary Fig. 3a). Knockdown of PB2 decreased the degradation of endogenous JAK1 after PR8 virus infection (Fig. 2f and Supplementary Fig. 2e). Moreover, we generated a PB2-mutant with a 5-nucleotide nonsense mutation in the target sequence of the shPB2 plasmids (ΔPB2) (Supplementary Fig. 3b, c). Both ΔPB2 and PB2 dramatically inhibited the expression of JAK1 in shGFP cells (Supplementary Fig. 3d), while ΔPB2, but not PB2, dramatically inhibited the expression of JAK1 in stable PB2 knockdown cells (Supplementary Fig. 3d). In addition, the degradation of JAK1 induced by CHX was more strongly in the presence of PB2 protein, indicating that PB2 decreased the stability of JAK1 protein (Fig. 2g and Supplementary Fig. 2f).

The ubiquitin-proteasome and autophagy-lysosome pathways are two different systems that control protein degradation in eukaryotic cells. We found that PB2-mediated JAK1 degradation could be mostly restored by treatment with the proteasome inhibitor MG132, but not the lysosome inhibitor ammonium chloride (NH4Cl), or chloroquine (CQ) (Fig. 2h and Supplementary Fig. 2g). Collectively, these results suggest that PB2 mediates the degradation of JAK1 through the proteasome machinery.

### IAV PB2 mediates polyubiquitination degradation of JAK1
As the ubiquitin system plays an important role in regulating protein degradation, we determined if PB2 promotes the ubiquitination of JAK1. As anticipated, we found that PB2 from PR8 or CZ potentiated JAK1 ubiquitination in a dose-dependent manner (Fig. 3a). Since different polyubiquitination processes, including K48-, K63-, K11-, and K27-linked ubiquitination, have been implicated in deliberately regulating protein fate. We set out to analyze which type of JAK1 ubiquitination could be induced by PB2. Overexpression of PB2 markedly promoted K48-linked ubiquitination of JAK1 but had no appreciable effect on the ubiquitination of JAK1 with other linkages (Fig. 3b and Supplementary Fig. 4a). We used reverse ubiquitin mutants to confirm these results, where only one lysine residue, K48, is mutated to

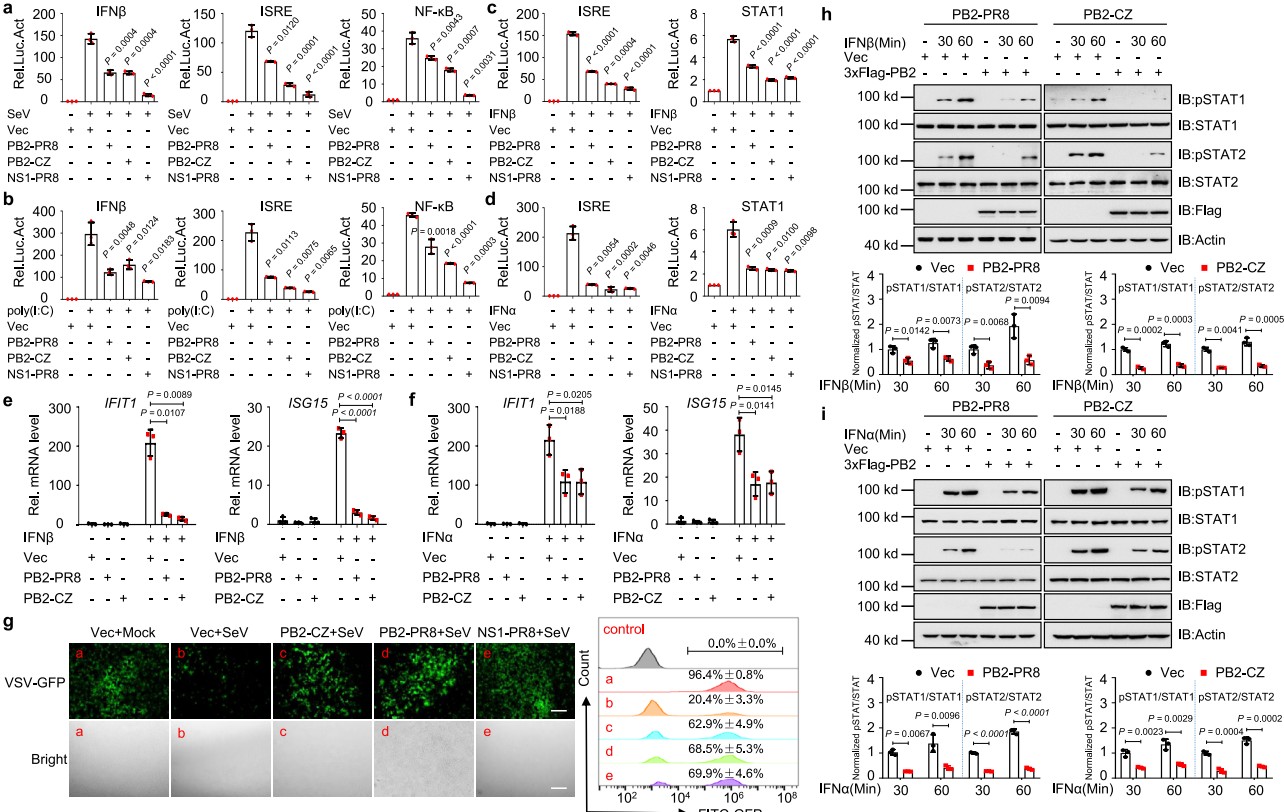

**Fig. 1 | IAV PB2 protein inhibits IFN signaling pathway. a, b** Luciferase activity in HEK293T cells transfected with IFNβ promoter, ISRE or NF-κB luciferase reporter, Renilla luciferase plasmid, and PB2 plasmids (including NS1-PR8 and empty vector [Vec] were chosen as positive and negative controls, respectively) and treated with SeV (**a**) or poly(I:C) (**b**) (*n* = 3 biologically independent samples). **c, d** Luciferase activity in HEK293T cells transfected with ISRE or STAT1 promoter-luciferase reporter plasmid, Renilla luciferase plasmid, and PB2 plasmids and treated with IFNβ (**c**) or IFNα (**d**) (*n* = 3 biologically independent samples). **e, f** qPCR analysis of *IFIT1* and *ISG15* mRNA in A549 cells transfected with PB2 plasmids and treated with IFNβ (**e**) or IFNα (**f**); mRNA results are presented relative to those of untreated cells transfected with a control plasmid (*n* = 3 biologically independent samples).

**g** HEK293T cells were transfected with PB2 or NS1 plasmid and infected with SeV. The supernatants were inactivated by ultraviolet radiation and collected to treat fresh HEK293T cells for 24 h, followed by infection for 12 h with VSV-GFP. The cells were observed under microscopy and then assessed by flow cytometry. Scale bars, 200 μm. **h, i** Immunoblot analysis of phosphorylated and total STAT1 or STAT2 in HEK293T cells transfected with PB2 plasmids and treated with IFNβ (**h**) or IFNα (**i**) (upper). Densitometry analysis of phosphorylated STAT/total STAT ratio (lower). Data are presented as the mean ± SD and are one representative of three independent experiments. Statistical significance in **a**–**f**, **h**, **i** was determined by unpaired two-tailed Student's *t* test.

---

arginine. The level of JAK1 ubiquitination was consistently lower upon expression of Ub-K48R, firmly establishing that the PB2 mediates K48-linked ubiquitination of JAK1 (Supplementary Fig. 4b). Moreover, we found that IAV infection enhanced the K48-linked ubiquitination of endogenous JAK1 (Fig. 3c). In contrast, knockdown of PB2 inhibited the IAV-induced K48-linked endogenous ubiquitination of JAK1 (Fig. 3d). These data suggest that PB2 promotes K48-linked ubiquitination and degradation of JAK1 by the proteasome pathway.

### The Lysine 859 and 860 residues of JAK1 are critical for efficient PB2-mediated degradation

The JAKs contain seven conserved, termed JAK homology (JH) regions[35]. The C-terminal half of the JAK, domains JH1 (amino acids 851-1154) and JH2 (amino acids 560-851), the JH1 domain, a classical tyrosine kinase domain, the pseudokinase domain (JH2) that may play a regulatory role[36,37]. The N-terminal half of the JAK1, domains JH3 to JH7, contains a predicted FERM domain and a putative SH2[35]. To identify the specific domain of JAK1 necessary for ubiquitination and degradation, we designed and constructed the following truncate plasmids: JAK1-1-559aa (amino acids 1-559), JAK1-560-1154aa (amino acids 560-1154), JAK1-1-850aa (amino acids 1-850), and JAK1-851-1154aa (amino acids 851-1154) (Fig. 4a). JAK1-WT, JAK1-851-1154aa, and JAK1-560-1154aa, but not JAK1-1-559aa and JAK1-1-850aa, were extensively degraded by PB2 (Fig. 4a). We next investigated which lysine residues

in JAK1 are targeted by PB2. To this end, we mutated a series of lysines of JAK1 (Fig. 4b) and examined their effects on PB2-mediated JAK1 degradation. As shown in Fig. 4c–e, only simultaneous mutations of Lysine 859 and Lysine 860 to Arginine (K859/860R) abolished PB2-mediated degradation of JAK1 (Fig. 4c–e). To further dissect the ubiquitination of JAK1 at K859/K860 residues, JAK1-WT, JAK1(K859R), or JAK1(K860R) was strongly ubiquitinated when HA-tagged wild-type ubiquitin was expressed, whereas ubiquitination of JAK1 (K859/860R) was significantly reduced (Fig. 4f, g). These data indicate that the Lysine 859 and 860 residues of JAK1 are critical for PB2-mediated degradation.

### Degradation of JAK1 via K859/K860 ubiquitination enhances IAV replication

To investigate the role of JAK1 degradation during IAV replication, we first determined if overexpression of JAK1 affects IAV replication. Infection experiments showed that transient overexpression of JAK1 in HEK293T cells reduced PR8 (Fig. 5a) or CZ (Fig. 5b) viral replication. To further understand the endogenous role of JAK1 during IAV infection, we employed shRNA-mediated knockdown of JAK1 expression (Fig. 5c). The results showed that viral titers were facilitated in JAK1-deficient cells (Fig. 5d, e). To verify the effects of JAK1 (K859/K860) on IAV replication, we generated a JAK1-mutant with a 7-nucleotide nonsense mutation in the target sequence of the shJAK1 plasmids (ΔJAK1)

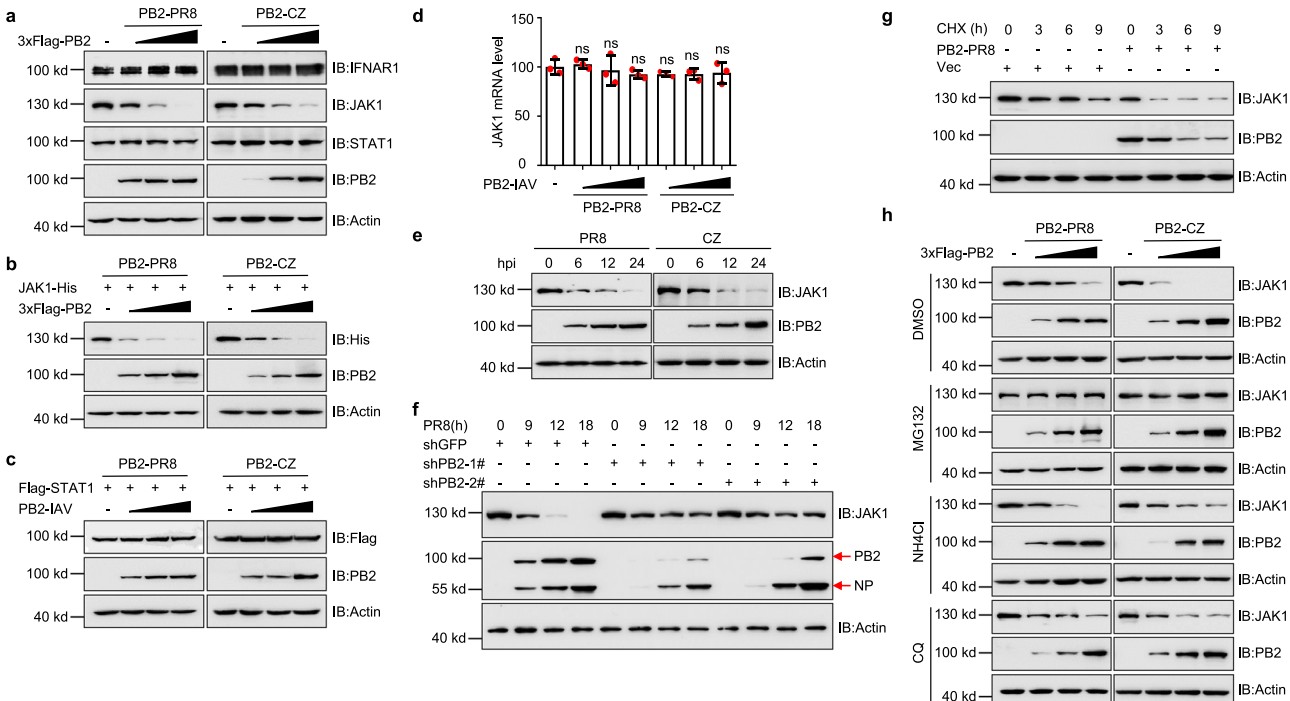

**Fig. 2 | IAV PB2 promotes the degradation of JAK1. a** Immunoblots of HEK293T cells were transfected with different concentrations of PB2 plasmids from PR8 (left) or CZ (right). **b, c** Immunoblots of HEK293T cells transfected with JAK1-His (**b**) or Flag-STAT1 (**c**) with different concentrations of PB2 plasmids. **d** qPCR analysis of *JAK1* mRNA level in A549 cells transfected with different concentrations of PB2 plasmids (*n* = 3 biologically independent samples). Data are presented as the mean ± SD. Statistical significance was determined by unpaired two-tailed Student's *t* test. [ns] *P* > 0.05. **e** Immunoblots of A549 cells infected with PR8 (left) or CZ (right) at an MOI of 1. **f** Immunoblots of stable PB2 knockdown HEK293T cells infected with PR8 virus (MOI = 0.1). **g** Immunoblots of HEK293T cells were transfected with PB2-PR8 and treated with CHX. **h** Immunoblots of HEK293T cells were transfected with the PB2 plasmids and treated with DMSO, MG132, NH4Cl, or chloroquine (CQ). Data are one representative of three independent experiments.

(Fig. 5f, g). Interestingly, overexpression of wild-type JAK1 but not the K859/860R mutants JAK1 reduced virus replication in shJAK1 cells, suggesting that mutants of K859/860R damaged JAK1-mediated antiviral activity (Fig. 5h, i). Altogether, our results indicate that JAK1 ubiquitination degradation at K859/K860 residues is critical for efficient IAV replication.

## JAK1 interacts with IAV PB2

We then determined whether JAK1 was associated with PB2. First, we examined if JAK1 physically interacts with PB2. Immunoprecipitation (IP) and Ni-NTA pull-down experiments were performed. IP 3xFlag-PB2 was able to bring down His-tagged JAK1, and 3xFlag-PB2 can be pulled down with His-tagged JAK1 (Fig. 6a), indicating a physical interaction of the two proteins. Second, as PB2 is associated with viral RNA, we further determined whether the interaction between PB2 and JAK1 is mediated by RNA. RNase A treatment was used to evaluate the effect of RNA on the binding of PB2 with JAK1. RNase A completely degraded RNA extracted from cells (Supplementary Fig. 5a). Regardless of RNase A treatment, 3xFlag-PB2 could be pulled down by the Ni-NTA from cells co-expressing JAK1-His (Supplementary Fig. 5b), suggesting RNA-independent interactions between PB2 and JAK1. Next, IP and Ni-NTA pull-down experiments further verified that PB2 was associated with JAK1 following IAV infection (Fig. 6b). Moreover, we examined the subcellular distribution of JAK1 upon influenza virus infection at 12 h. JAK1 was evenly distributed in the whole-cell cytosol in mock-infected cells (Fig. 6c). At 12 h post-infection (p.i), JAK1 was present predominantly in the cytoplasm and co-localized with PB2 protein to form punctate bodies (Fig. 6c).

IAV PB2 protein is comprised of five major domains: N-terminal PB1 Binding Domain (PBD, amino acids 1–240), Middle Domain (MD, amino acids 241–322), Cap Binding Domain (CBD, amino acids

323–485), RNA Binding Domain (RBD, amino acids 486–532) and C-terminal Nuclear import Domain (NID, amino acids 533–759) (Fig. 6d). We conducted IP and Ni-NTA pull-down experiments and confirmed that the CBD and RBD, but not the PBD, MD, and NID are the interacting domain of PB2 for binding to JAK1 (Fig. 6e, f).

In an attempt to map the region of JAK1 responsible for interaction with PB2, we designed and constructed the following truncate plasmids: JAK1-301–1154aa (amino acids 301–1154), JAK1-1-435aa (amino acids 1–435), JAK1-436-1154aa (amino acids 436–1154), JAK1-1-559aa (amino acids 1–559), JAK1-1-850aa (amino acids 1–850), and JAK1-851-1154aa (amino acids 851–1154) (Supplementary Fig. 5c). Deletion of the kinase-like and Kinase domains (amino acids 560-1154) completely abolished the binding of JAK1 to PB2, whereas JAK1 with any other mutant domains had a binding affinity similar to that of full-length JAK1 (Supplementary Fig. 5d), indicating that the kinase-like and kinase domains of JAK1 are the interacting domains for binding to PB2, which differs from the SH2 domains (amino acids 436–560) of JAK1 for cytokine receptor binding[38,39]. Altogether, these data indicate that JAK1 interacts with IAV PB2.

## PB2s of H5 subtype AIVs degrade mammalian JAK1 differently

A previous study identified 283M/526R of PB2 from CZ as a virulence marker in mice compared to A/duck/Eastern China/JY/2014 (JY)[19]. To investigate the role of PB2s of different H5 subtype AIVs on JAK1 degradation, HEK293T cells were transfected with JAK1-His and PB2M (M283I/R526K)-CZ, PB2-CZ, PB2-JY, or PB2M (I283M/K526R)-JY. Intriguingly, we found that the PB2 from JY displays no degradation of JAK1 (Fig. 7a). Notably, PB2-CZ or PB2M-JY, but not PB2M-CZ, induced the degradation of JAK1 (Fig. 7a). In addition, we observed PB2-CZ, PB2-CZ-M283I, and PB2-CZ-R526K, but not PB2M-CZ downregulate JAK1 protein levels (Fig. 7b), indicating a synergistic effect of PB2 283M and

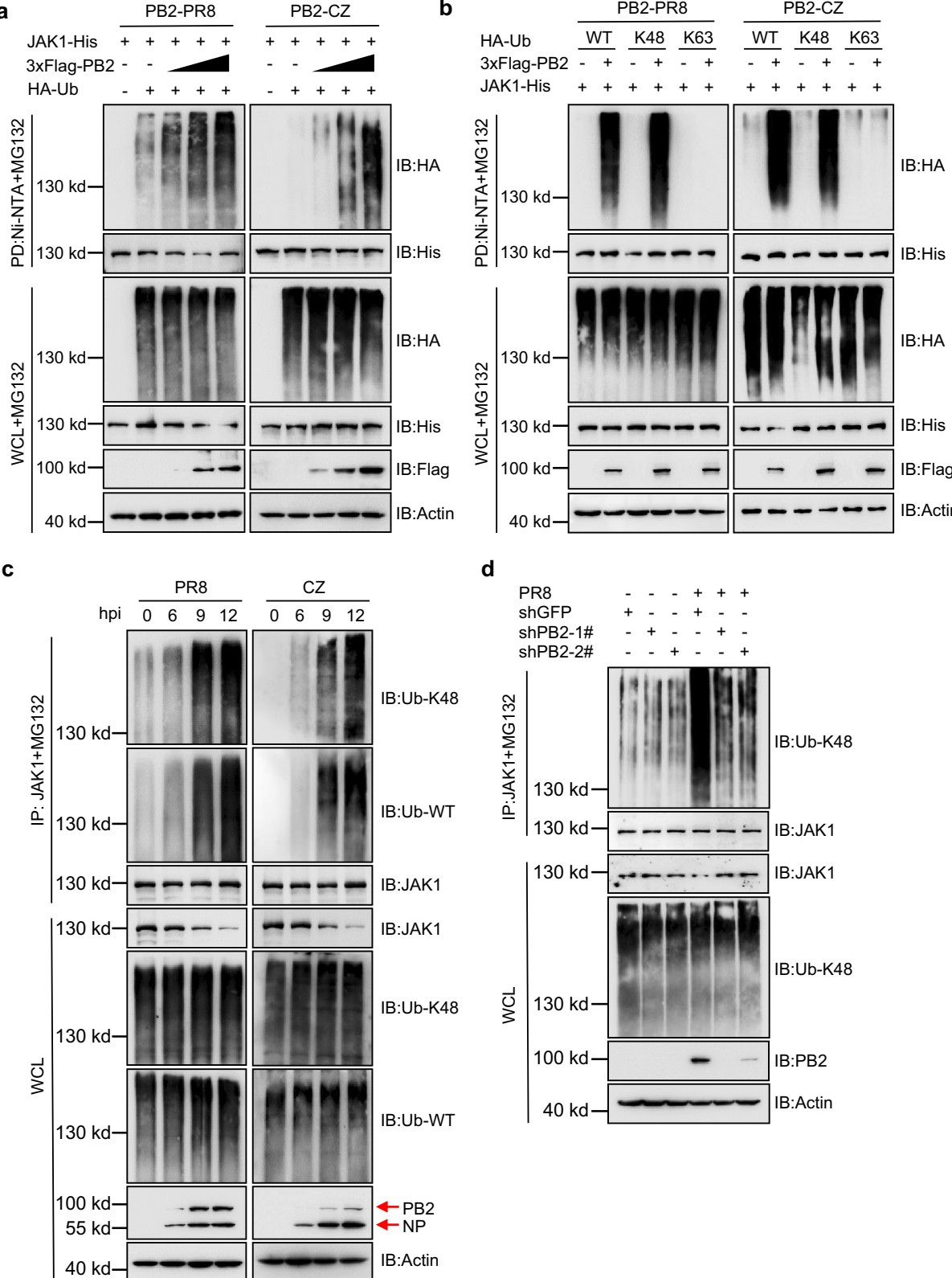

**Fig. 3 | IAV PB2 protein increases the K48-linked ubiquitination of JAK1. a** Ni-NTA pull-down analysis of the ubiquitination of JAK1 in HEK293T cells transfected with JAK1-His, HA-ubiquitin (HA-Ub), and PB2 plasmids and treated with MG132. **b** Ni-NTA pull-down analysis of the ubiquitination of JAK1 in HEK293T cells transfected with JAK1-His, HA-Ub, or its mutants K48 and K63 [K at indicated residue, and K at other residues were simultaneously mutated to arginines], and PB2 plasmids and treated with MG132. **c** Co-IP analysis of the ubiquitination of JAK1 in A549 cells infected with PR8 or CZ at an MOI of 0.01. **d** Co-ip analysis of the ubiquitination of JAK1 in stable PB2 knockdown HEK293T cells infected with PR8 virus. WCL, whole-cell lysates. Data are one representative of three independent experiments.

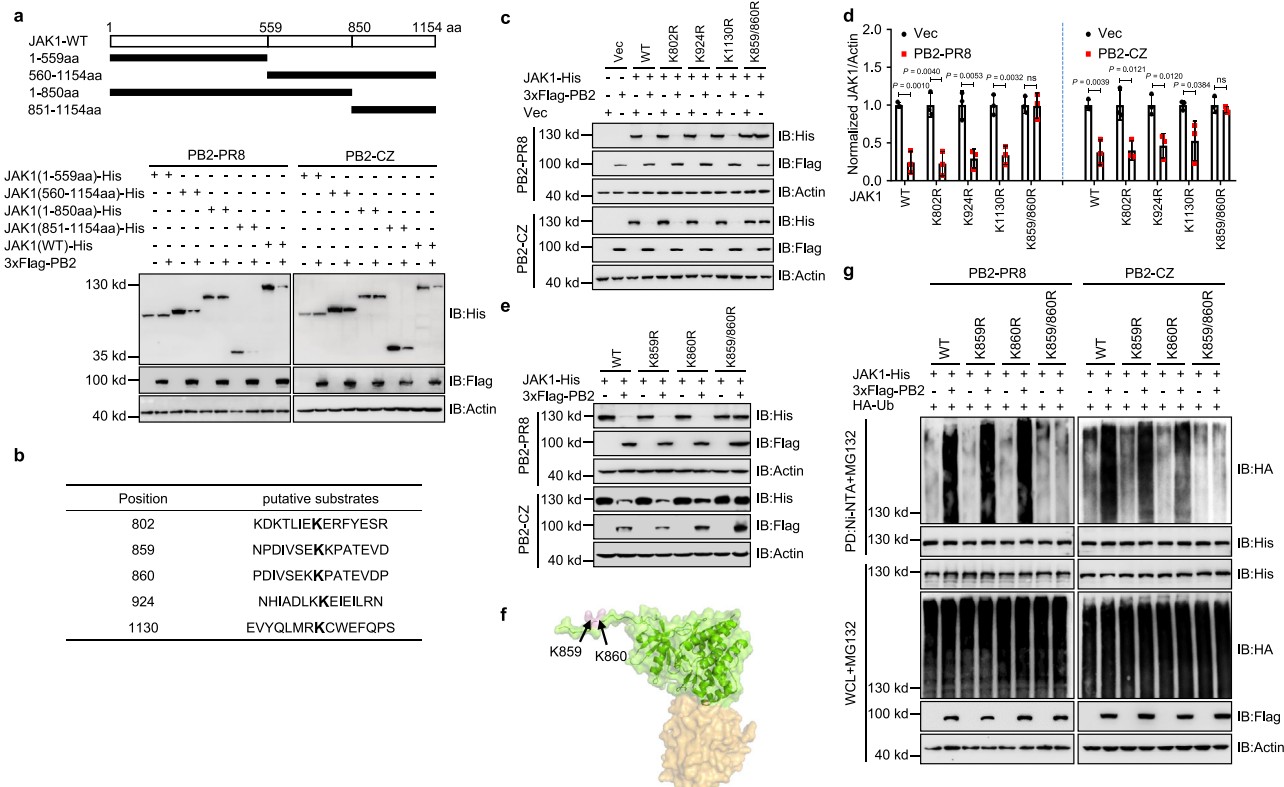

**Fig. 4 | PB2-mediated JAK1 degradation at residues 859K and 860K. a** Schematic representation of the deletion mutants of JAK1 (upper) and immunoblots of HEK293T cells transfected with JAK1 or its deletion mutants and PB2 plasmids (lower). **b** Ubiquitination modification online prediction of JAK1 by CPLM 1.0. **c** Immunoblots of HEK293T cells transfected with JAK1 or its mutants and PB2 plasmids. **d** Intensities analysis of the bands on the immunoblots (c) from three independent experiments. Data are presented as the mean ± SD and statistical significance was determined by unpaired two-tailed Student's *t* test. ˢ *P* > 0.05. **e** Immunoblots of HEK293T cells transfected with JAK1 or its mutants and PB2 plasmids. **f** Molecular model of the JAK1 kinase domain generated by PyMOL (PDB: 4ehz). Residues K859/860 are highlighted in pink. **g** Ni-NTA pull-down analysis of the ubiquitination of JAK1 in HEK293T cells transfected with JAK1 or its mutants, HA-Ub and PB2 plasmids, and treated with MG132. WCL, whole-cell lysates. Data are one representative of three independent experiments.

526R contributes to JAK1 degradation. To investigate the effect of 283M and 526R of PB2 on JAK1 interaction. IP assay revealed that the JAK1 protein is bound with PB2-CZ more than PB2-JY (Fig. 7c). The PB2M-CZ substantially decreased its binding to JAK1 compared with PB2-CZ, whereas the PB2M-JY showed the opposite results (Fig. 7c).

Furthermore, we examined whether IAV PB2 co-localizes with JAK1 in the same cell compartments by fluorescence microscopy. As anticipated, PB2-JY is consistently localized primarily in the nucleus, as PB2 contains a nuclear localization sequence (Fig. 7d). However, JAK1 did not co-localize with PB2-JY (Fig. 7d). Then, we used recombinant viruses with double substitutions (I283M/K526R) in the PB2 and tested their localization. JAK1 co-localized with PB2 appeared in CZ-infected cells (Fig. 6c), but this colocalization was not observed in double substitutions (PB2-M283I/R526K-CZ) in the rCZM-infected cells (Fig. 7d). In contrast, this colocalization was observed in double substitutions (PB2-I283M/K526R-JY) in the rJYM-infected cells (Fig. 7d).

Although most PB2 protein in infected cells localizes to the nucleus, PB2 was also detected in the mitochondria[40,41]. To assess the effect of 283M/526R on PB2 proteins intracellular distribution, A549 cells were infected with rCZ, rJY, rCZM, and rJYM, respectively. At 6 and 12 hpi, mitochondria were labeled with MitoTracker Red, and PB2 proteins were detected via indirect immunofluorescence. The PB2 protein from the rCZ virus displayed significant levels of the mitochondrial signal at 12 hpi, as expected (Supplementary Fig. 6). Surprisingly, we also found that PB2 proteins from the other three H5N8 viruses, in contrast with the rCZ virus, do not associate with mitochondria (Supplementary Fig. 6). Interestingly, PB2-JY and PB2M-CZ proteins localized primarily to the nucleus in infected cells, while PB2-

CZ and PB2M-JY can also be observed in the cytoplasm. To further understand whether the PB2-JAK1 interaction affects the ubiquitination degradation of JAK1. The results showed that PB2-CZ or PB2M-JY, but not PB2-JY or PB2M-CZ promoted the K48-linked ubiquitination of JAK1 (Fig. 7e).

The Lysine 859 and 860 residues of JAK1 are critical for efficient ubiquitination and degradation of JAK1 mediated by PB2-CZ, which are also highly conserved among various species (Supplementary Fig. 7a). We tested the ability of PB2 from CZ or JY to interact with chicken JAK1 (chJAK1). Immunoblotting and IP experiments showed that PB2-CZ and PB2-JY efficiently interacted with chJAK1 for degradation (Supplementary Fig. 7b, c). Moreover, we observed that both PB2-CZ and PB2-JY promoted the ubiquitination of chJAK1 (Supplementary Fig. 7d). These results collectively demonstrate that the PB2s of H5 subtype AIVs are different in mammalian JAK1 degradation, not in avian JAK1 degradation.

## AIV PB2 inhibits the JAK1/STAT1 signaling pathway by degrading JAK1

Given that the protein level of JAK1 could affect cellular sensitivity to IFNs and the PB2 proteins from H5 AIVs has different ability to degrade mammalian JAK1, we sought to gain insight into the regulatory function of PB2s from different H5 AIVs in the IFNs mediated signaling pathway. First, the transcription profiling of ISGs such as *IFIT1*, *ISG15*, and *TAP1* in response to IFNβ in the presence or absence of AIV PB2 was analyzed by qPCR. PB2-CZ and PB2M-JY, but not PB2M-CZ and PB2-JY, were shown to substantially inhibit IFNs-triggered transcription of *IFIT1*, *ISG15*, and *TAP1* (Fig. 8a, b). Second, the promoter activity of the

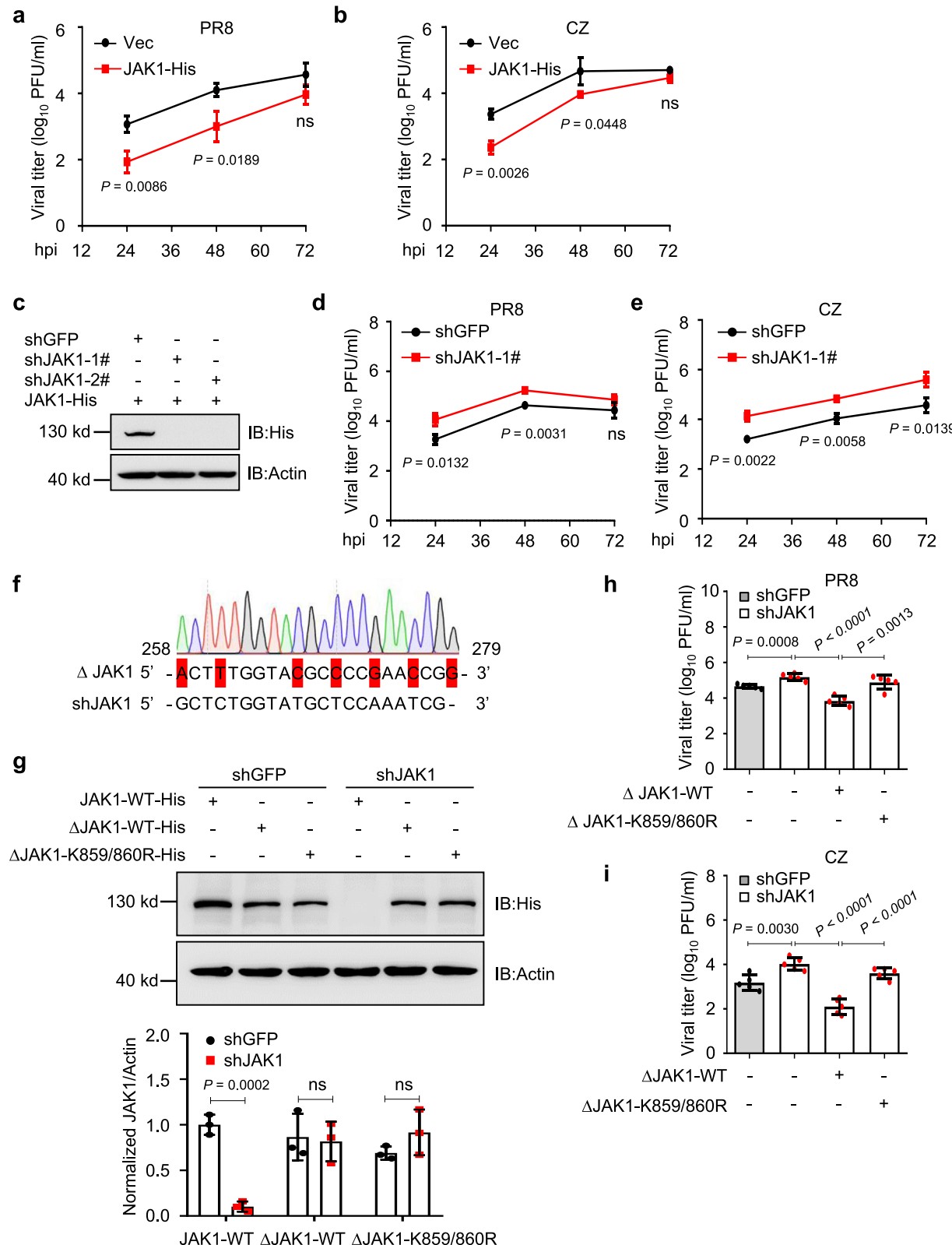

ISRE and STAT1 in response to IFNs was measured by luciferase reporter assay. Following the treatment with IFNβ and IFNα, the ISRE and STAT1 promoter's relative activities were markedly suppressed by the expression of PB2-CZ and PB2M-JY, but not PB2M-CZ and PB2-JY (Fig. 8c, d).

We next investigated the activation of STAT1 in response to IFNs in the presence or absence of PB2 expression. Exogenous addition of

IFNs elicited STAT1 activation, as evidenced by the heightened phosphorylation level of STAT1. However, PB2-CZ and PB2M-JY, not PB2M-CZ and PB2-JY expression, inhibited the IFNβ (Fig. 8e) or IFNα (Fig. 8f) induced pSTAT1/pSTAT2. Since PB2 degrades JAK1 at lysine 859 and 860 residues, we investigated the effect of JAK1 (K859/860R) on the IFN-induced expression of ISGs. The results revealed that IFNβ-induced levels of *ISG15, IFIT1*, and *TAP1* mRNA were enhanced only in

**Fig. 5 | The effects of JAK1 on IAV replication. a**, **b** Viral titers in HEK293T cells transfected with JAK1 plasmid and infected with PR8 virus (**a**) or CZ virus (**b**) ($n = 3$ biologically independent samples). **c** Immunoblots of HEK293T cells transfected with control shRNA (shGFP) or shRNA targeting *JAK1* (shJAK1) and JAK1 plasmids. **d**, **e** Viral titers in shJAK1 HEK293T cells infected with PR8 virus (**d**) or CZ virus (**e**) ($n = 3$ biologically independent samples). hpi, h post-infection. **f** JAK1 was replaced with an shRNA off-target JAK1 mutant (ΔJAK1) with a 7-nucleotide nonsense mutation in the target sequence of the shJAK1 plasmid. **g** Immunoblots of stable

JAK1 knockdown HEK293T cells transfected with JAK1-His, ΔJAK1-His, or ΔJAK1-K859/860R-His plasmid (upper). Intensities of the bands on the immunoblots from three independent experiments were quantified and normalized with actin (lower). **h**, **i** Viral titers in shJAK1 HEK293T cells transfected with ΔJAK1-WT or ΔJAK1-K859/860R plasmid and infected with PR8 virus (**h**) or CZ virus (**i**) ($n = 5$ biologically independent samples). Data are presented as the mean ± SD and are one representative of three independent experiments. Statistical significance in **a**, **b**, **d**, **e**, **g**–**i** was determined by unpaired two-tailed Student's *t* test. [ns] $P > 0.05$.

the shJAK1 cells transfected with JAK1(WT)-expressing plasmid, not in those transfected with JAK1 (K859/860 R)-expressing plasmid (Fig. 8g), suggesting the mutations of K859/860R reduce JAK1-mediated antiviral immune response. These results indicate that PB2-mediated JAK1 degradation robustly decreases cellular sensitivity to IFN.

## AIV PB2-mediated JAK1 degradation is critical for virus transcription and replication in vitro

To further identify the effect of JAK1 degradation by AIV PB2 on virus replication. A549 cells were infected with these recombinant AIVs. The expression levels of viral proteins PB2 and NP were significantly decreased in rCZM virus-infected cells compared to rCZ infected cells (Fig. 9a). Moreover, the viral proteins were significantly increased in rJYM virus-infected cells compared to rJY-infected cells (Fig. 9b). Meanwhile, rCZM infection-induced less JAK1 degradation than rCZ infection (Fig. 9a). Consistently, rJYM infection-induced more JAK1 degradation than rJY infection (Fig. 9b), suggesting the PB2 degrades JAK1 to benefit IAV replication. Further, we compared the rCZ, rJY, rCZM, and rJYM viruses' growth curves in mammalian-origin A549 cells and avian-origin DF-1 cells. In A549 cells, the virus titer of rCZ was higher than that of rCZM at 12 hpi, and the growth of rCZ was faster than that of rCZM ($10^{6.5}$ PFU/ml vs. $10^{4.5}$ PFU/ml at 48 hpi), indicating that the M283I/R526K mutation of PB2 weakened the replication ability of rCZ (Fig. 9c left). In contrast, the rJYM virus displayed more replication than the parental rJY virus observed in virus titers at late time points ($10^{5.3}$ PFU/ml vs. $10^{4.2}$ PFU/ml at 72 hpi) (Fig. 9c right). While all of these viruses grew similarly and efficiently in DF-1 cells and reached the maximum titers of approximately $10^{7.3}$ TCID50/mL at 72 hpi (Fig. 9d). Importantly, the size of the plaques formed by rCZM in MDCK cells was significantly smaller than those formed by rCZ as measured by mean diameter (0.74 mm vs. 1.31 mm, in Fig. 9e) and the size of the plaques formed by rJYM in MDCK cells was bigger than those formed by rJY as measured by mean diameter (0.80 mm vs. 0.10 mm, in Fig. 9e).

Next, we examined the levels of viral mRNA, cRNA, and vRNA in A549 cells or DF-1 cells infected with rCZ, rJY, rCZM, or rJYM at 12 hpi. The results showed that A549 cells (Supplementary Fig. 8a–c), but not DF-1 cells (Supplementary Fig. 8d–f) infected with viruses carrying 283M/526R PB2 have significantly higher levels of viral nucleoprotein (NP) mRNA, cRNA, and vRNA, as compared with cells infected with a virus-containing 283I/526K PB2 (Supplementary Fig. 8), suggesting 283M/526R are critical for viral transcription and replication in mammalian cells. Altogether, these data indicate that the AIV PB2-mediated JAK1 degradation benefits virus replication in mammalian cells.

## AIV PB2-mediated JAK1 degradation significantly increases viral pathogenicity in mice

Since downregulation of JAK1 is critical for efficient IAV replication in vitro, it is most probable that the level of JAK1 degradation by PB2 could be related to viral pathogenicity. To test this hypothesis, we intranasally infected 4–6 weeks mice with rCZ, rJY, rCZM, and rJYM viruses and monitored their weight loss and survival for two weeks. The mice infected with rCZM exhibited lower body weight loss and morbidity than infection with rCZ (Fig. 10a, b). Consistently, the mice

infected with rJYM exhibited more significant weight loss and a higher level of morbidity than those in the rJY group (Fig. 10a, b).

Notably, severe pneumonia was observed via gross and histopathological lesions in the lung tissues of mice inoculated with rJYM and rCZ viruses on 2 and 5 days post-infection (dpi), characterized by consolidation, hemorrhages, and edema (Fig. 10c). In contrast, little or no lung consolidation was observed in mice infected with rJY and rCZM virus at 2 dpi (Fig. 10c). Compared with the rCZ group, the virus titer in the lung of rCZM group was lower (Fig. 10d). Of note, in lung tissue, mice infected with rJYM displayed a higher viral titer than rJY (Fig. 10d). Furthermore, histopathological assessment of lungs from the infected mice showed that rCZ caused serious bronchiolitis and obvious inflammatory cell infiltration on 2 dpi, and progressed to bronchopneumonia on 5 dpi (Fig. 10e, f). In contrast, rCZM-infected mice showed moderate pathogenic changes on 5 dpi (Fig. 10e, f). Consistently, rJYM-infected mice showed the extent of bronchiolitis was markedly increased compared with rJY on 5 dpi (Fig. 10e, f). In addition, viral protein NP staining was more intense in lung sections of mice infected with rJYM or rCZ than in corresponding mice infected with rJY or rCZM (Fig. 10g, h).

Lung tissues of infected mice were examined further to determine the relationship between viral infection and JAK1. Consistently, the expression of JAK1 was lower in the lungs of mice infected with rJYM or rCZ than that of mice infected with rJY or rCZM (Fig. 10i, j). We also analyzed the levels of ISGs mRNA in the lungs of mice infected with the variant or the WT (Fig. 10k, l). On 2 dpi, the levels of *ISG15* (Fig. 10k) and *IFIT1* (Fig. 10l) mRNA in the mice infected with the rJY or rCZM were higher than the levels in the mice infected with the rJYM or rCZ. Altogether, these results demonstrated that PB2-mediated JAK1 degradation significantly increases the virulence of AIV in mice.

## Discussion

IAVs are among the most common contagious pathogens to cause severe respiratory infections. Host cells have developed multiple branches of the innate immune system to defend against IAV infection, including activation of the IFNs system. However, IAVs also developed diverse strategies to antagonize the IFNs system for better infection. This study identified IAV PB2 protein as a negative regulator of JAK1 to attenuate the IFNs-induced antiviral response. IAV PB2 mediates polyubiquitination and degradation of JAK1 at lysines 859 and 860, which contribute to viral replication in mammalian cells and pathogenicity in a mouse model.

The virulence of an influenza virus depends on many factors, such as cell tropism, replicate efficiency, and host immunity. Innate immunity plays a key role in host defense against IAV infection. The IFNs are secreted and bind to the cognate receptor to activate the JAK/STAT signal pathway, thereby inducing the expression of ISGs, which include *Mx1*[42], *ISG15*[43], and *TAP1*[44], that can repress IAV replication. The influenza virus has devised diverse strategies to interfere with producing IFNs and ISGs. For example, influenza viral proteins such as NS1[45], PB2[46], and PB1-F2[47] were reported to have the ability to disrupt the signaling pathways of IFN synthesis. The influenza viral protein NS1 inhibits IFN production by inhibiting the ubiquitin E3 ligase TRIM25-mediated K63-linked ubiquitination of the viral RNA sensor RIG-I[32,33],

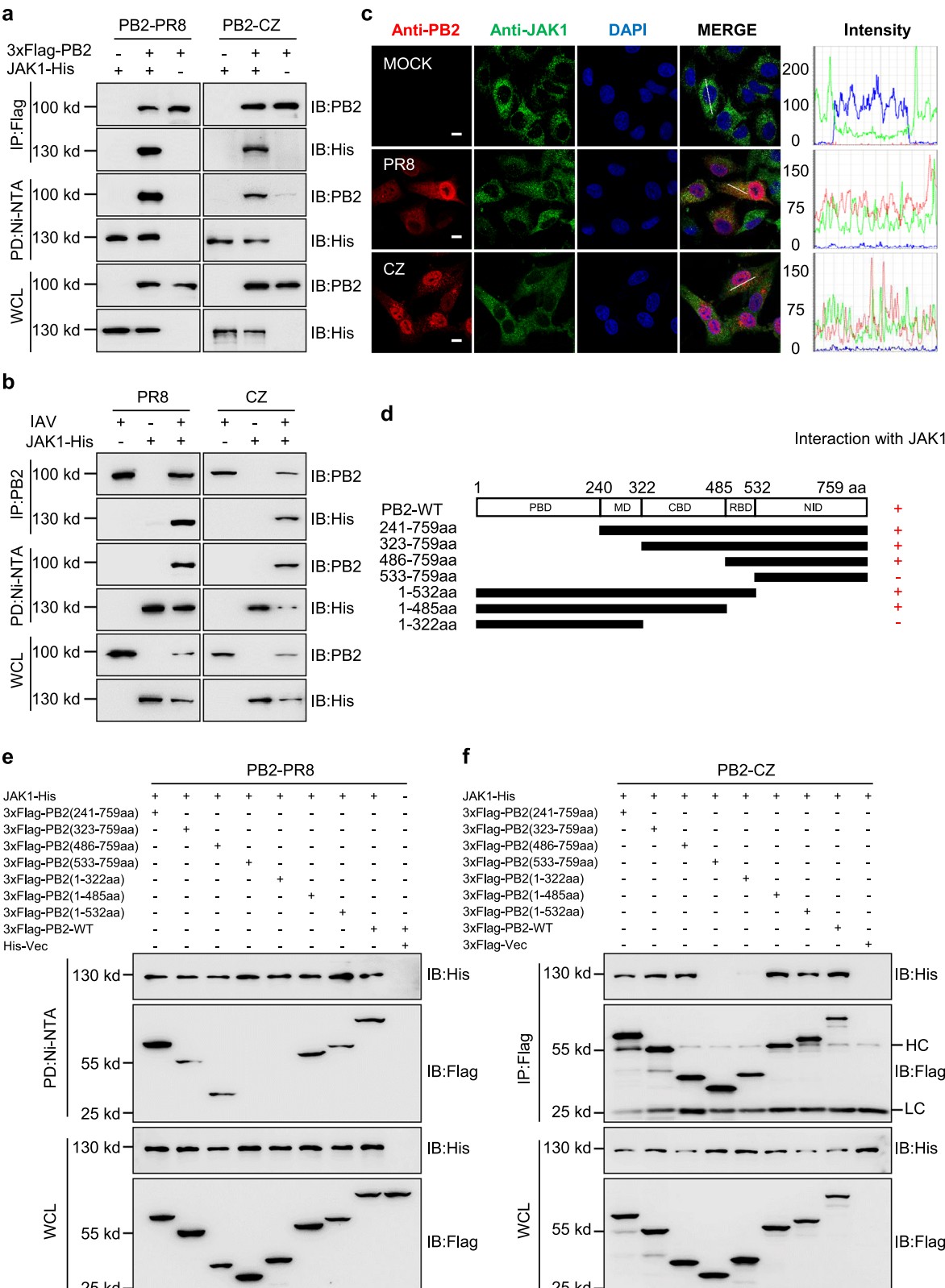

**Fig. 6 | JAK1 interacts with IAV PB2. a** Co-ip and Ni-NTA pull-down analysis of the interaction of PB2 with JAK1 in HEK293T cells transfected PB2 and JAK1 plasmids. **b** Co-ip and Ni-NTA pull-down analysis of the interaction of PB2 with JAK1 in HEK293T cells transfected with JAK1 plasmids and infected with PR8 or CZ virus. **c** Colocalization of endogenous JAK1 (green) and PB2 (red) in PR8 or CZ-infected A549 cells. Nuclei were stained with DAPI (blue). Scale bars, 10 μm. Intensities of fluorescence at indicated locations were scanned by LAS X Software. **d** Schematic representation of the deletion mutants of PB2. **e**, **f** Ni-NTA pull-down (**e**) and Co-ip analysis (**f**) of the interaction of JAK1 with PB2 and its truncation mutants in HEK293T cells. WCL whole-cell lysates, HC Heavy Chain, LC Light Chain. Data are one representative of three independent experiments.

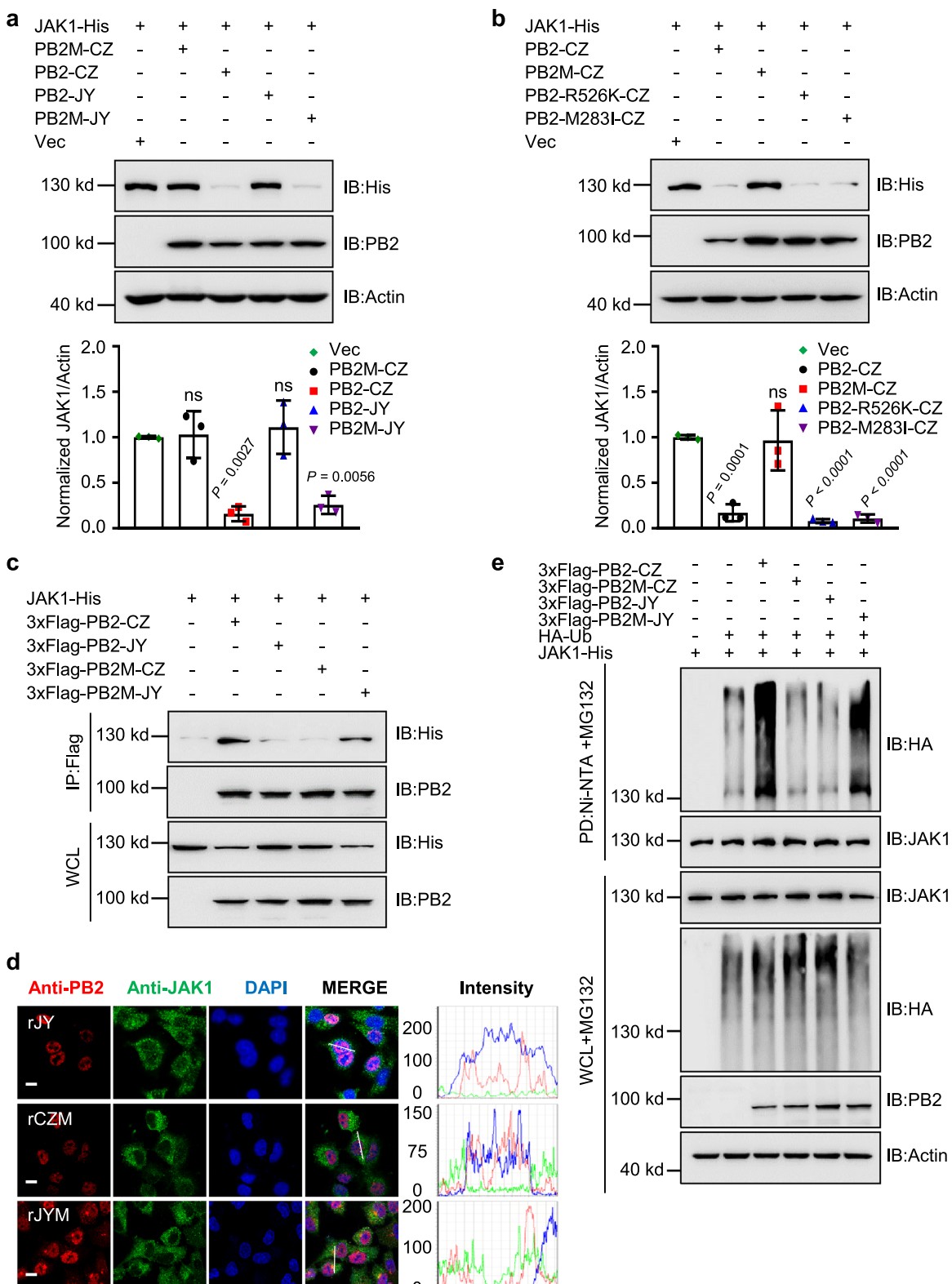

while influenza NS1 protein targets TRIM25 in a species-specific manner for the inhibition of RIG-I ubiquitination and antiviral IFN production[48]. Recent research also shows that NS1 binds to interferon-inducible protein IFI35 in a strain-specific manner, NS1 encoded by swine (H3N2), but not avian (H7N9/2013), interacts with IFI35, hence affecting RIG-I mediated innate antiviral response[49]. In addition, NS1 was shown to regulate JAK/STAT signaling by binding to and degrading the DNA methyltransferase 3B (DNMT3B)[50]. Previous research has

reported that PB2 can interact with the mitochondrial antiviral signaling protein (MAVS) to disrupt IFN induction[46]. The present study demonstrates that PB2 protein attenuated cellular responses to IFNs by degrading JAK1. This strategy could potently repress JAK1/STAT signaling, creating a cellular environment favorable for the replication and propagation of the influenza virus.

Many RNA viruses or viral proteins prohibit STAT1/2 activation from hindering ISGs production. SARS-CoV-2 nsp1 and nsp6

**Fig. 7 | PB2s of H5 subtype AIVs degrade mammalian JAK1 differently.**
**a** Immunoblots of HEK293T cells transfected with JAK1 and PB2-CZ, PB2M-CZ (M283I-R526K), PB2-JY, or PB2M-JY (I283M-K526R) plasmids (upper). The intensities of the bands on the immunoblots from three independent experiments were quantified and normalized with actin (lower). Data are presented as the mean ± SD.
**b** Immunoblots of HEK293T cells transfected with JAK1 and PB2-CZ, PB2-R526K-CZ, PB2-M283I-CZ, or PB2M-CZ plasmids (upper). The intensities of the bands on the immunoblots from three independent experiments were quantified and normalized with actin (lower). Data are presented as the mean ± SD. **c** Co-ip analysis of the

interaction of PB2 or its mutants with JAK1 in HEK293T cells. **d** Colocalization of endogenous JAK1 (green) and PB2 (red) in rJY, rCZM (M283I-R526K), or rJYM (I283M-K526R) infected A549 cells. Nuclei were stained with DAPI (blue). Scale bars, 10 μm. Intensities of fluorescence at indicated locations were scanned by LAS X Software. **e** Ni-NTA pull-down analysis of the ubiquitination of JAK1 in HEK293T cells transfected with PB2 or its mutant plasmids and treated with MG132. WCL, whole-cell lysates. Statistical significance in **a**, **b** was determined by unpaired two-tailed Student's *t* test. ns *P* > 0.05. Data are one representative of three independent experiments.

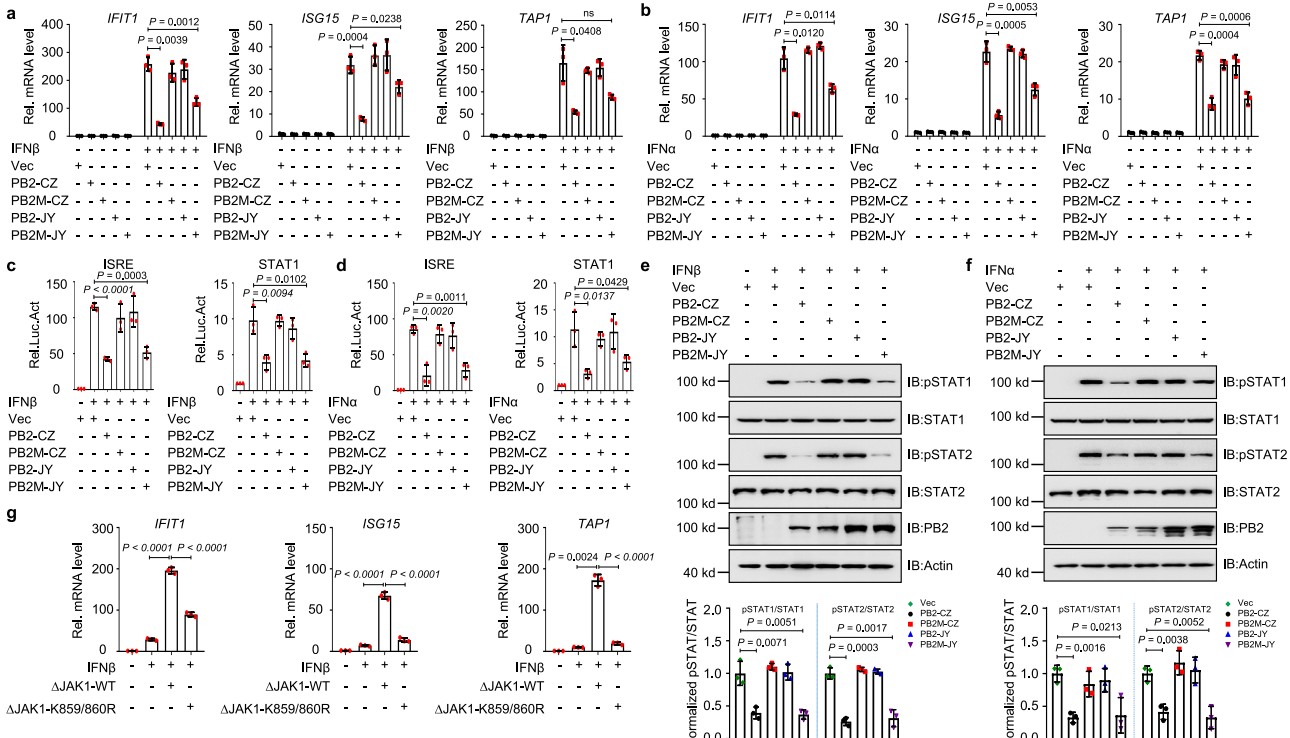

**Fig. 8 | AIV PB2 inhibits IFN-mediated signaling by JAK1 degradation. a**, **b** qPCR analysis of *IFIT1*, *ISG15*, and *TAP1* mRNA in A549 cells transfected with PB2 or its mutant plasmids and treated with IFNβ (**a**) or IFNα (**b**) (*n* = 3 biologically independent samples). **c**, **d** Luciferase activity in HEK293T cells transfected with ISRE or STAT1 promoter-luciferase reporter, Renilla luciferase plasmid, and PB2 or its mutants plasmids and treated with IFNβ (**c**) or IFNα (**d**) (*n* = 3 biologically independent samples). **e**, **f** Immunoblot analysis of phosphorylated and total STAT in

HEK293T cells transfected with PB2 or its mutant plasmids and treated with IFNβ (**e**) or IFNα (**f**) (upper). Densitometry analysis of the ratio of phospho-STAT/total STAT on the immunoblots (lower). **g** qPCR analysis of *IFIT1*, *ISG15*, and *TAP1* mRNA in shJAK1 HEK293T cells transfected with ΔJAK1-WT or ΔJAK1-K859/860R plasmid and treated with IFNβ. Data are presented as the mean ± SD and are one representative of three independent experiments. Statistical significance was determined by unpaired two-tailed Student's *t* test. ns *P* > 0.05.

antagonize IFN-I signaling through blocking STAT1/STAT2 phosphorylation or nuclear translocation[51]. The V proteins of the Nipah virus and Hendra virus block IFNβ and IFNγ signaling pathways by preventing the phosphorylation and nuclear accumulation of STAT1[52,53]. However, the V protein of the Measles virus blocks IFNα/β but not IFNγ signaling by inhibiting STAT1/STAT2 phosphorylation[54]. In addition, FMDV VP3 prevents IFN-induced STAT1-activated tyrosine phosphorylation and blocks the nuclear accumulation of phosphorylated STAT1[25]. In the present study, we revealed that PB2 weakened the activation of STAT1/STAT2 and reduced the phosphorylation of STAT1/STAT2 by targeting JAK1 for degradation. Our detailed biochemical analysis revealed that the C-terminal JH2 and JH1 domains of JAK1 are associated with CBD and RBD of PB2. JH2 domain of JAK1 is an active tyrosine kinase, while the JH1 domain of JAK1 is a pseudokinase domain that regulates both basal and cytokine-induced activation. However, the JH2 domain of JAK1 is the primary site for PB2-mediated degradation, and the Lys 859 and 860 residues in the JH2 domain of JAK1 are critical for efficient PB2-mediated degradation (Fig. 4). In addition, we found that the PB2

protein of H9N2/TX, H7N9/GD, H5N1/YZ, and H5N1/DT influenza viruses also promote the degradation of JAK1 (Supplementary Fig. 9), suggesting a general degradation JAK1 effect of PB2 from different influenza virus subtypes.

The ubiquitin-proteasome system (UPS) is a key player in signal transduction pathways, including innate immune responses. Ubiquitylation modification is increasingly recognized as a key strategy used by viral pathogens to modulate host factors critical for infection. K48-linked polyubiquitin chains play a vital role in protein degradation by targeting proteins to the proteasome. IAV can use the UPS to evade the host's immune system. IAV HA expression leads to IFNAR1 ubiquitination and degradation to attenuate the IFN-induced antiviral signaling pathway[34]. The viral NS1 protein, a well-known virulence factor, developed many mechanisms to antagonize innate immunity, from direct interactions with UPS factors, such as TRIM25, OUTB1, and MDM2, to the general inhibition of host gene expression[29,32,55,56]. Ubiquitination assay confirmed that PB2 catalyzed the formation of K48-linked polyubiquitin chains attached to JAK1 at lysine 859 and 860

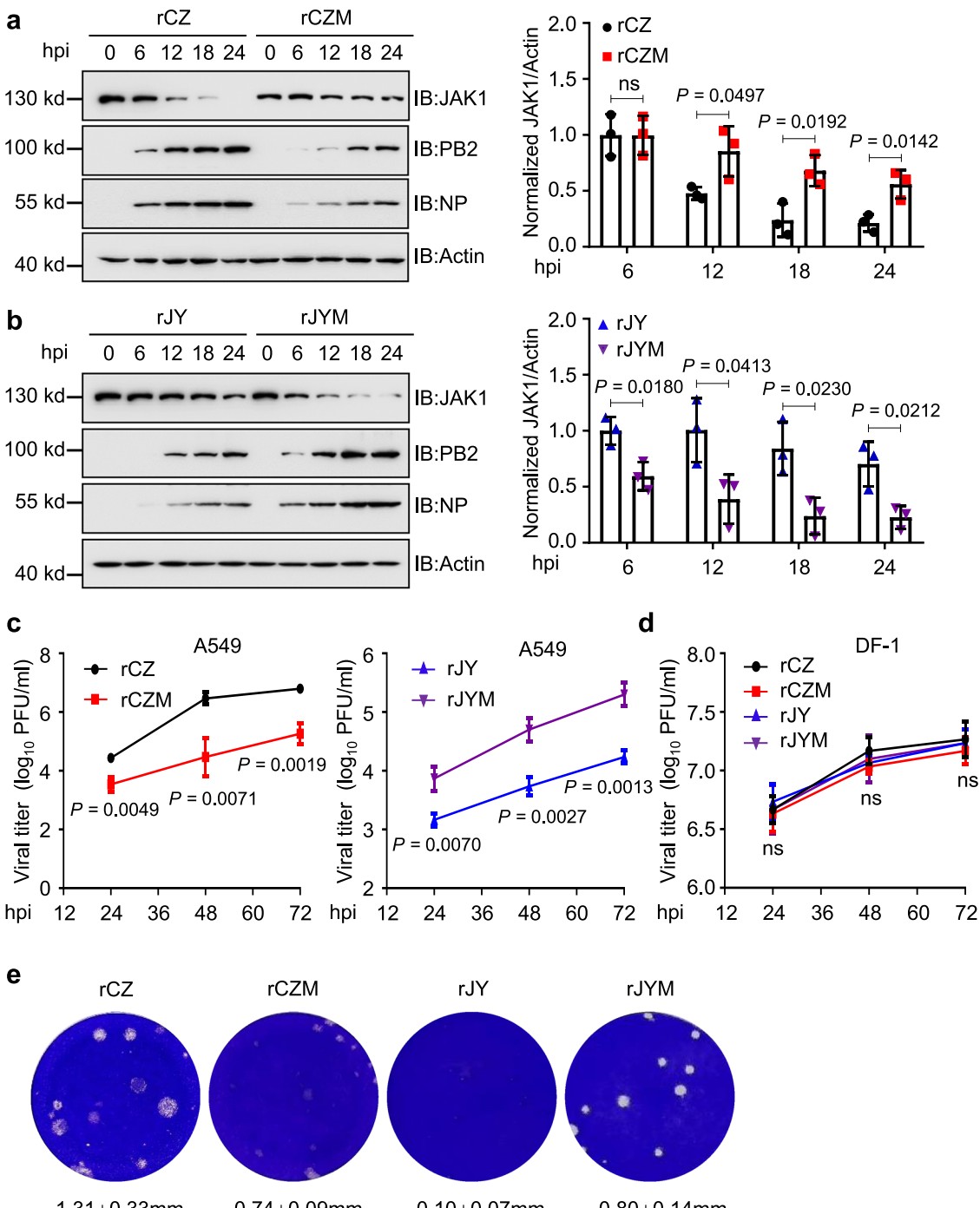

**Fig. 9 | JAK1 degradation mediated by AIV PB2 affects virus replication in mammalian cells. a, b** Immunoblots of A549 cells infected with rCZ, rCZM, rJY, or rJYM virus (left). The intensities of the bands on the immunoblots from three independent experiments were quantified and normalized with actin (right). **c, d** Growth curve of viruses in A549 (**c**) or DF-1 cells (**d**) infected with rCZ, rCZM, rJY, or rJYM at an MOI = 0.01 (*n* = 3 biologically independent samples). hpi, h postinfection. **e** The plaque morphology of MDCK cells infected with rCZ, rCZM, rJY, or rJYM virus. Data are presented as the mean ± SD. Statistical significance was determined by unpaired two-tailed Student's *t* test. ns *P* > 0.05.

residues. As expected, PB2 markedly downregulated the mRNA abundance of IFN-stimulated cytokines, including *ISG15*, *IFIT1*, and *TAP1*, in response to IFNs treatment. It is conceivable that IAV PB2 causes JAK1 degradation, which helps the virus escape the powerful IFNs system, which may correlate with viral pathogenesis. Interestingly, although mutation of K859/860R of JAK1 led to its degradation resistance, these mutations also weakened JAK1-mediated antiviral immune response (Fig. 8g), which could explain why reconstitution of the JAK1 K859/860R mutants did not restore the antiviral activity of

JAK1 in the shJAK1 cells (Fig. 5h, i). These data suggest that K859/860 is the essential site for both JAK1 ubiquitination and JAK1-mediated ISGs antiviral activity.

Host adaptation requires the virus to overcome restriction barriers and replicate efficiently in a new host. Given the complicated replication process and life cycle of IAV, there are multiple host constraints for different stages of viral replication. For instance, RIG-I directly binds to incoming nucleocapsids, with enhanced specificity for PB2-627E RNP complexes, following the viral genome release from

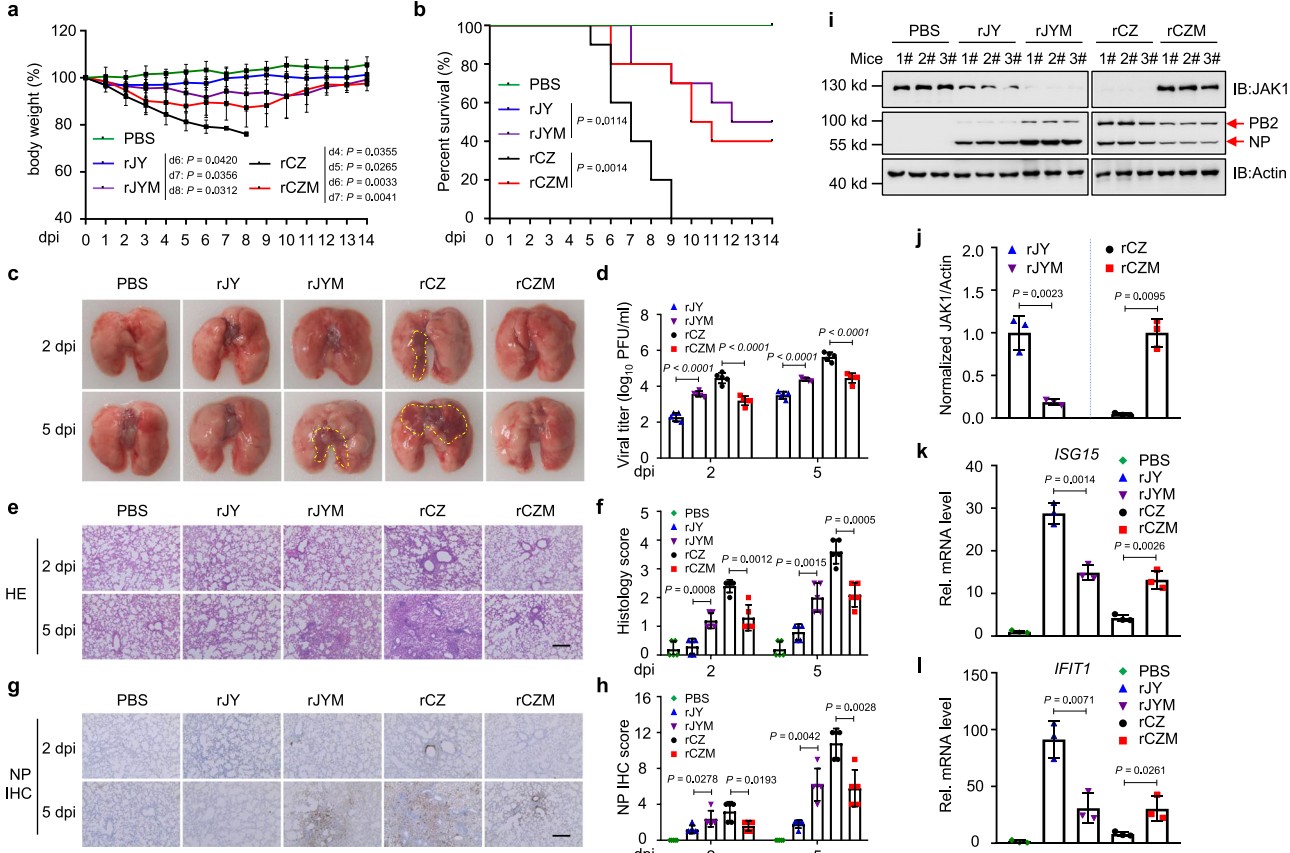

**Fig. 10 | JAK1 degradation mediated by AIV PB2 significantly increases the virulence of AIV in mice. a, b** The mice ($n = 10$ per group) were intranasally inoculated ($10^6$ $EID_{50}$ per mouse) with rJY, rJYM, rCZ, or rCZM virus and monitored for 14 d for body weight loss (**a**) and survival (**b**). **c** Gloss lesions of lungs from the infected mice on 2 and 5 dpi. Severe pneumonia with diffuse consolidation was outlined in yellow. The images are from representative one of five mice. **d** Viral titers in the lungs (n = 5 per group) were determined on 2 and 5 dpi by plaque assay. **e**–**h** Hematoxylin/eosin (HE) staining (**e**) and scoring (**f**), immunohistochemistry (IHC) staining (**g**), and scoring of lung sections (**h**). Scale bar, 200 μm. The images are from representative one of five mice. The scores were calculated from five mice. **i** Immunoblots of lung tissue from the infected mice. **j** Quantification of JAK1 expression on the immunoblots (**i**) and normalized with actin ($n = 3$). **k, l** qPCR analysis of *ISG15* (**k**) and *IFIT1* (**l**) mRNA in lung from the infected mice on 2 dpi ($n = 3$). dpi, days post-infection. Data are presented as the mean ± SD. Statistical significance was determined by unpaired two-tailed Student's *t* test in **a, d, f, h, j**–**l** or *log-rank* test in **b**.

endosomes during influenza virus infection[57]. Nuclear import is another barrier that determines the replication efficiency of the influenza virus and is associated with PB2 host adaptation[58]. The effect of PB2 host adaptations on viral polymerase activity has been extensively studied, with the host factor ANP32A recently being found to restrict viral polymerase activity[17]. Here, we found that the PB2s of H5 subtype AIVs had a different ability for JAK1 degradation in mammalian cells, consisting of virus replication in mammalian cells and virus pathogenicity in mice. The rJY strain formed much smaller plaques in MDCK cells than the rCZ strain, and its growth kinetics were also severely compromised in A549 cells due to defective in degrading JAK1 (Fig. 9). Consistent with the in vitro growth properties, the recombinant viruses with weak JAK1 degradation ability and strong JAK1-mediated ISGs expression exhibited lower replicate efficiency in mouse lung tissue and caused less weight loss and mortality in infected mice (Fig. 10). Therefore, the ability of AIV PB2 for JAK1 degradation is related to its virulence in mammalian.

In summary, we propose a working model for the regulatory role of the PB2 protein of IAV in the JAK1-mediated signaling pathway. IAV PB2 protein blocks JAK1/STAT signaling by targeting JAK1 for degradation through the proteasome machinery (Supplementary Fig. 10). IAV PB2 protein-mediated JAK1 degradation is critical for inhibiting antiviral IFNs response, which enhances understanding of the viral immune evasion mechanism. The ability of PB2 protein-mediated JAK1 degradation is also identified as a molecular marker for AIVs

adaptation to mammals, which facilitates the surveillance of AIVs for cross-species transmission.

## Methods

### Ethics statement
Animal experiments were conducted following guidelines for experimental animals' welfare and ethics. All animal studies follow the protocols of Jiangsu Province Administrative Committee for Laboratory Animals (approval number: SYXK-SU-2017-0044). According to the institutional biosafety manual, all experiments with virus infection were carried out at a biosafety level 3 Laboratory and animal facility at Yangzhou University (CNAS BL0015).

### Cell culture and virus infection
Human embryonic kidneys (HEK) 293T cells (ATCC; CRL-11268), the chicken fibroblast cell line DF-1 cells (ATCC; CRL-12203) and human lung epithelial A549 cells (ATCC; CCL-185) were cultured in Dulbecco's modified Eagle's medium (DMEM; HyClone, USA). MDCK cells (ATCC; CCL-34) were cultured in minimum essential medium Eagle (MEM; HyClone, USA). Cells were routinely cultured at 37 °C with 5% CO2, and all media were supplemented with 10% fetal bovine serum. A/Puerto Rico/8/34 (PR8) was identified and stored in our laboratory[59]. Two HPAI H5N8 viruses used in this study, A/goose/Eastern China/CZ/2013 (CZ) and A/duck/Eastern China /JY/2014 (JY), have been characterized in our previous studies[60]. Recombinant viruses were rescued by

reverse genetics using eight plasmid-based reverse genetic systems described previously[61]. Eight segments from the CZ and JY strains were cloned into bidirectional reverse-genetics plasmid pHW2000 as described before[19]. The CZ or JY backbone was used to rescue recombinant viruses with various PB2 genes containing different mutants. The rCZM was rescued with the CZ system of the PB2-CZ in the double substitutions (M283I-R526K). The rJYM was rescued with the JY system of the PB2-JY in the double substitutions (I283M/K526R)[19].

Cells were inoculated with IAV at the indicated multiplicity of infection (MOI) for virus infection. After adsorption for 1 h at 37 °C, the cells were washed with phosphate-buffered saline (PBS) and cultured in DMEM containing 0.5 μg/ml TPCK (L-1-tosylamido-2-phenylethyl chloromethyl ketone) treated trypsin. For in vivo infection, 4-6 week-old female BALB/c mice (Yangzhou Experimental Animal Center, Yangzhou, China) were intranasally inoculated with $10^{6.0}$ $EID_{50}$ of each indicated virus. Animals were observed daily for mortality, and body weight was measured up to 14 d post-infection (p.i). On 2 and 5 dpi, five mice from each group infected with virus-containing rJY, rJYM, rCZ, rCZM, or mock-infected with PBS were euthanized, and lung tissues were collected from each mouse for virus titration, immunoblotting, histochemical staining and real-time quantitative PCR (RT-qPCR).

All mice were raised in individually ventilated cages in the animal facilities of Yangzhou University. They were housed in a controlled environment with a 12 h light/12 h dark cycle and kept with free access to food and water throughout the whole experiment period. Room temperature was maintained at 25 °C and humidity level was controlled between 40–60%.

## Titrations
Viral replication kinetics. A549 or DF-1 cells were inoculated with an MOI of 0.01. The supernatant was harvested at 24, 48, and 72 hpi. MDCK cells were inoculated with 10-fold serial dilutions of viruses for the plaque assay. The cells were washed with phosphate-buffered saline (PBS) 1 h after inoculation and replaced with 2 × DMEM mixed with an equal portion of 1% agarose containing 0.5 μg/ml TPCK treated trypsin. After a 72 h incubation, MDCK cells were fixed with 4% formaldehyde and stained with crystal violet.

## Reagents and antibodies
Proteasome inhibitor MG132 (Beyotime Biotechnology), lysosome inhibitor NH4Cl (Fisher Scientific), Chloroquine (CQ, Sigma-Aldrich), 4′, 6-diamidino-2-phenylindole dihydrochloride (DAPI, Sigma-Aldrich), Protein A/G PLUS-Agarose (Santa Cruz). Dual-Luciferase Reporter Assay System (Promega and Vazyme), protease inhibitor phenylmethylsulfonyl fluoride (PMSF, Beyotime Biotechnology), Cycloheximide (CHX, Sigma-Aldrich), recombinant human IFNβ and IFNα (GenScript), anti-Flag M2 affinity agarose (Sigma-Aldrich) were purchased from the indicated manufacturers. Mouse anti-NP mAb (GTX629633, 1:1000) and rabbit anti-PB2 pAb (GTX125926, 1:1000) were purchased from GeneTex. Rabbit anti-STAT1 mAb (14995S, 1:1000), rabbit anti-pSTAT1 mAb (8826S, 1:1000), rabbit anti-STAT2 mAb (72604S, 1:1000), rabbit anti-pSTAT2 mAb (88410S, 1:1000), mouse anti-Ubiquitin (Ub) mAb (3936S, 1:500), rabbit anti-K48-linkage specific polyubiquitin mAb (8081S, 1:500), mouse anti-HA tag mAb (2367S, 1:500), rabbit anti-JAK1 mAb (29261S, 1:1000) and mouse anti-JAK1 mAb (50996S, 1:1000) were purchased from Cell Signaling Technology. Rabbit anti-beta Actin pAb (Abcam, ab8227, 1:1000); mouse anti-beta Actin mAb (Santa Cruz, sc47778, 1:2000), mouse anti-JAK1 mAb (Zen BIO, 200622-8B8, 1:500), rabbit anti-Interferon alpha/beta receptor 1 pAb (Abcam, ab245367, 1:500), mouse anti-His tag mAb (HUABIO, M0812-3, 1:2000), rabbit anti-Ub mAb (HUABIO, ET1609-21, 1:500) and mouse anti-Flag M2 mAb (Sigma-Aldrich, F1804, 1:2000) were purchased from the indicated manufacturers. Alexa Fluor™ 488

Goat anti-Mouse IgG (H + L) (A11029, 1:400), Alexa Fluor™ 594 Goat anti-Rabbit IgG (H + L) (A11037, 1:400) and Alexa Fluor™ 488 Goat anti-Rabbit IgG (H + L) (A11034, 1:400) were purchased from Invitrogen.

## Plasmid construction and transfection
Standard molecular biology procedures were performed for all plasmid constructions. The full-length cDNA encoding JAK1 cloned into the pCDNA3.1-His. To produce various truncated forms of the plasmids JAK1, indicated fragments of DNA were amplified from JAK1-His. Several deletion forms of the JAK1 plasmid were produced and designed as JAK1(301-1154aa)-His, JAK1(1-435aa)-His, JAK1(436-1154aa)-His, JAK1(1-559aa)-His, JAK1(560-1154aa)-His, JAK1(1-850aa)-His and JAK1(851-1154aa)-His. Viral genes were amplified from the IAV genomes by RT-PCR and were cloned into 3xFlag-CMV-10 or pHW2000. PB2-H9N2/TX was cloned from A/chicken/Taixing/10/2010 (H9N2/TX), PB2-H7N9/GD was cloned from A/chicken/Guangdong/4/2017 (H7N9/GD), PB2-H5N1/YZ was cloned from A/chicken/Yangzhou/11/2016 (YZ/H5N1), PB2-H5N1/DT was cloned from A/chicken/Jiangsu/DT1/2016 (H5N1/DT), PB2-CZ was cloned from CZ, PB2-JY was cloned from JY and PB2-PR8 was cloned from PR8. Several deletion forms of PB2 plasmid were produced and designed as 3xFlag-PB2(241-759aa), 3xFlag-PB2(323-759aa), 3xFlag-PB2(486-759aa), 3xFlag-PB2(533-759aa), 3xFlag-PB2(1-532aa), 3xFlag-PB2(1-485aa) and 3xFlag-PB2(1-322aa). Plasmids for HA-Ub or its mutants and Flag-STAT1 were constructed by standard molecular biology techniques. A Mut Express® II Fast Mutagenesis Kit V2 (Vazyme) was used to generate the JAK1, Ub, or PB2 gene mutations. The IFN-β-Luc, NF-κB-Luc, STAT1-Luc, ISRE-Luc, and pRL-TK internal control luciferase reporter plasmids used in the study were described previously[62,63]. Primer sequences used for cloning are available upon request. According to the manufacturer's recommendations for transient expression, all constructs were validated by sequencing and were transfected into cells using Transfection Reagent (BioBEST).

## RNA interference
The sequences of the 2 vector-based shRNAs targeting different regions of PB2 or JAK1 mRNA transcript were: 5′-GCTGTGACATGGT GGAATAGG-3′ (shPB2-1#), and 5′-GCTAAAGCATGGAACCTTTGG-3′ (shPB2-2#); 5′-GCTCTGGTATGCTCCAAATCG-3′ (shJAK1-1#), and 5′-GGTGGAAGTGATCTTCTATCT-3′ (shJAK1-2#). An shRNA targeting enhanced green fluorescent protein (EGFP) was used as a negative control. The shRNA-mediated in vitro screening assay was used to generate the stable specific knockdown cell lines[64,65]. Briefly, HEK293T cells were cotransfected with shRNAs expressed in plasmids against the PB2 or JAK1 and the corresponding plasmids expressing the full-length PB2 or JAK1. The interference efficiency of each shRNA clone in silencing the expressed relevant protein was assessed by immunoblotting with indicated antibody. The efficiency shRNA was then packaged into a lentivirus to infect HEK293T cells. After puromycin selection, the stable HEK293T cell line with the downregulated PB2 or JAK1 was established after continuous selection for 3 weeks, when the control mock-infected HEK293T cells died out completely.

## RT-qPCR analysis
According to the manufacturer's instructions, total RNA was purified using Trizol Reagent (Sigma-Aldrich) and was treated with DNase I (Thermo Scientific) to remove contaminated DNAs. First-strand complementary DNA was synthesized from 1 μg of total RNA using a TransScript RT reagent kit (Thermo Scientific). Uni-12, Uni-13, and oligo dT primers were used for reverse transcription of vRNA, cRNA, and mRNA, respectively. Oligo dT and random primers were used for detecting host genes. Generated cDNA was subjected to qPCR in a 20 μl reaction volume using gene-specific primers. cDNA quantities were normalized to the GAPDH/Actin. qPCR primers used in this study are provided in Supplementary Table 1.

## VSV-GFP bioassay

Antiviral cytokine secretion bioassays were conducted as previously described, with slight modifications[66]. Briefly, HEK293T cells were transfected with the indicated plasmids. At 24 h post-transfection (hpt), the cells were infected with SeV (MOI = 1) for another 24 h. The supernatants were harvested and inactivated by placing the samples on ice with a 30-W ultraviolet radiation lamp for 20 min. The ultraviolet radiation-inactivated supernatant was added to fresh confluent cells and incubated for 24 h. The cells were then infected at an MOI of 0.1 with VSV-GFP. At 12 hpi, VSV-GFP replication was visualized by monitoring the GFP expression level by fluorescence microscopy or flow cytometry.

## Confocal immunofluorescence assay

Coverslip adhered A549 cell monolayers were infected with IAV for indicated times. Treated cells were washed twice with Phosphate Buffered Saline-Tween 20 (PBST), fixed with cold acetone/methanol (1/1) for 20 min, and then air-dried. The fixed cells were incubated with primary antibodies overnight. Cells were incubated with secondary antibodies conjugated to Alexa Fluor™ 488 or Alexa Fluor™ 594. at 37 °C for 1 h. Cellular nuclei were stained with DAPI for 20 min. The triply-stained cells were washed thrice with PBS. Images were obtained with a Leica SP8 confocal microscope and analyzed using LAS X Software (V3.7.4).

## Immunoblotting and Co-immunoprecipitation (co-IP)

Samples were prepared and treated as described above, and cell monolayers were washed with PBS and lysed with lysis buffer containing 50 mM Tris-HCl (pH7.5), 150 mM NaCl, 5 mM EDTA, 0.5% NP-40, and an anti-protease Halt Protease Inhibitor Single-Use Cocktail (Promega). The protein concentrations were determined by the BCA analysis kit. Protein fractions were used to load an SDS-PAGE gel to perform immunoblotting. The co-IP assay was conducted as previously described with minor modifications[49]. Briefly, HEK293T cells transfected alone or cotransfected with indicated plasmids or A549 cells infected with IAV were lysed with NP-40 lysis buffer after 36 h. The supernatant was incubated with indicated antibodies at 4 °C for 2 h. Immune complexes were precipitated by incubating Protein A/G PLUS-Agarose (Santa Cruz, Dallas, TX) for 6 h at 4 °C. After three stringent washes in NP-40 lysis buffer, immunoprecipitated proteins were analyzed by immunoblotting.

## Ni-NTA pull-down assays

For Ni-NTA, pull-down assays were done as described[67]. HEK293T cells cultured in 6-cm plates were transfected with the indicated plasmids. 36 hpt, cells from each plate were collected and divided into two aliquots. One aliquot was lysed in lysis buffer and analyzed by immunoblotting to examine the expression of transfected proteins. Another aliquot was lysed in buffer A (6 M guanidine-HCl, 0.1 M $Na_2HPO_4/NaH_2PO_4$, 10 mM Tris-Cl, pH 8.0, 5 mM imidazole, and 10 mM β-mercaptoethanol) and subjected to sonication for a total of 30 s. Cell lysates were incubated with 40 μL of pre-equilibrated Ni-NTA beads (QIAGEN) overnight at 4 °C. The beads were washed 4 times sequentially with buffers A, B (8 M urea, 0.1 M $Na_2HPO_4/NaH_2PO_4$, 10 mM Tris-HCl, pH 8.0, and 10 mM β-mercaptoethanol), and C (same as B except pH 6.3), respectively. Beads with bound proteins were then boiled in an SDS sample buffer with 200 mM imidazole and were subjected to immunoblot.

## Luciferase assay

HEK293T cells were transfected with a mixture of luciferase reporter (IFNβ-Luc, ISRE-Luc, NF-κB-Luc, or STAT1-Luc), pRL-TK (renilla luciferase plasmid), together with indicated plasmids. 30 hpt, the cells were treated with SeV, poly(I:C), IFNβ, or IFNα for another 12 h. The cells were lysed with passive lysis buffer, and the luciferase activity in the lysates was determined with the Dual-Luciferase Reporter Assay System. The Renilla luciferase construct pRL-TK was simultaneously transfected as an internal control.

## Lung tissue histology

The trachea and lungs from control or virus-infected mice at indicated post-infection times were dissected, fixed in 10% formalin, embedded into paraffin, sectioned, stained with Hematoxylin and eosin (H&E) solution, and visualized by light microscopy for histologic changes. The lung histological scores were measured by a pathologist in a blinded manner following a standardized score system as previously described[68,69]. Three random areas were scored in each lung tissue sample, and the mean value was calculated. The histology score is the median value from five mice. Scores: 0, no damage; 1, mild damage; 2, moderate damage; 3, marked damage; and 4, severe histological changes; an increment of 0.5 was used if the levels of damage fell between two integers. For IHC assays, tissue sections were stained using an anti-influenza virus NP monoclonal antibody. The staining intensity was evaluated as 0, negative; 1, mild, 2, moderate, and 3, strong. Depending on the percentage of positive cells, the proportion score of NP expression was classified as follows: 0, 0%; 1, ≤10%; 2, 11–50%; 3, 51–80%; and 4, ≥81%. The total score was calculated by multiplying the intensity score by the proportion score[70].

## Statistical analysis

All statistical analyses were performed using GraphPad Prism software V8.0.1. Data are presented as the mean ± SD unless otherwise indicated. Differences between experimental and control groups were determined by the Student's $t$ test. For all tests, differences between groups were considered significant when the $P$-value was >0.05 (ns), <0.05 (*) and <0.01 (**). The Kaplan–Meier method was adopted for animal survival analysis to generate graphs, and the survival curves were analyzed with log-rank analysis.

## Reporting summary

Further information on research design is available in the Nature Research Reporting Summary linked to this article.

## Data availability

All relevant data are available from the authors. The data used to generate the image in Fig. 4f is available from PBD entry 4ehz (http://www.rcsb.org/structure/4ehz). Source data are provided with this paper.

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

## Acknowledgements

This study was supported by the National Key R&D Project (2021YFD1800202 to D.P.), the National Natural Science Foundation of China (31872473 to D.P., 31872477 to S.C., 32172942 to T.Q.), the Jiangsu Provincial Natural Science Fund for Excellent Young Scholars (BK20200105 to T.Q.), Jiangsu Province University Outstanding Science and Technology Innovation Team Project [(2021) NO.1] to D.P., the Postgraduate Research&Practice Innovation Program of Jiangsu Pro-vince (KYCX22_3535 to H.Y.), the Agricultural Science and Technology Independent Innovation Fund of Jiangsu Province [CX(22)3004] to S.C., the 2020 Interdisciplinary Project of Yangzhou University Veterinary Special Zone (yzuxk202004) to T.Q., the Priority Academic Program Development of Jiangsu Higher Education (PAPD) to D.P., and the Jiangsu Qinglan Project [(2021) NO.11] to S.C.

## Author contributions

D.P. and H.Y. conceived and designed the study. H.Y., Y.D., Y.B., N.X., Y.W., F.Y., and Y.D. performed the experiments. D.P., H.Y., S.C., and T.Q. analyzed the data. D.P., S.C., and X.L. provided critical resources. D.P. and H.Y. wrote the manuscript. All authors have contributed to, reviewed, and approved the manuscript.

## Competing interests

The authors declare no competing interests.
