## [Peer Review File · Nature Communications]

The influenza virus PB2 protein evades antiviral innate immunity by inhibiting JAK1/STAT signallingREVIEWER COMMENTS

Reviewer #1 (Remarks to the Author):

In "Influenza A virus Polymerase Protein PB2 Evades Antiviral Immunity by Inhibiting JAK1/STAT Signaling", Yang and colleagues show that PB2 can antagonize interferon signaling by mediating JAK1 degradation via the proteasome. The researchers further show that this function of PB2 increases influenza virus replication *in vitro* and *in vivo*. They identify the regions of JAK1 and PB2 that interact with each other and also identify the lysines in JAK1 that are ubiquitinated. A particularly interesting finding in this manuscript was that PB2s of H5 avian influenza viruses with residues 283I and 526K could degrade chicken JAK1 but not mammalian JAK1 and that this negatively impacted the replication of these viruses in mice and mammalian cells.

Overall, this manuscript is thorough and presents a lot of new findings. The experiments are well controlled, and the story is clear.

My only requirement is that the abstract be modified to better represent the findings of the paper. The authors should mention the mammalian versus avian adaptation results. There are also some spelling and grammatical errors that need to be corrected.

Reviewer #2 (Remarks to the Author):

In the current study, the authors investigate the interactions between influenza virus PB2 and the host cell JAK1 proteins. They demonstrate that PB2 inhibits the innate immune response by binding to and ubiquitinating JAK1. They explore the mechanism of this process and the differences between avian and mammalian viruses.

Major comments:

1. Having demonstrated the impact on innate responses *in vitro*, there is no *in vivo* data on this effect – the disease is more severe, but is IFN down or ISG in the lungs, could be measured by RT-PCR.
2. How many times were the *in vitro* studies repeated?
3. Figure 11 is not a clear demonstration of the mechanism of action. It has more dedicated to how JAK/Stat works than how PB2 inhibits it. A reworking of this figure, showing how the PB2 ubiquitinates the JAK (through what domain) and then how it tags it for proteasome.
4. The language throughout could be more clear, for example describing the degradation resistant proteins (line 247-249) is very unclear and needed multiple re-reads to understand.

Minor comments:

1. The figure legends are long and difficult to read because they include conclusions for each subpanel within them. Please make them more concise.
2. In figure 3, what are up and down panels – do you mean upper and lower?
3. What does WCL stand for in the figures, please include an explanation
4. Why are the UBIQ plots shown at high and low exposure?
5. In figure 4C, can you put a hyphen in the K859R/K860R double mutant to indicate it is a double mutant. Why was the K859R or K860R single mutants not tested in the study in panel C.
6. Are the K48, K63 mutations in ubiquitin, it is not clear.
7. Be consistent on SeV in the legend and the figure (in the figure it is put sev)

Reviewer #3 (Remarks to the Author):

In this very thorough biochemical study, Yang et al describe a novel mechanism by which Influenza A polymerase protein PB2 suppresses antiviral responses by directly targeting JAK1 to promote its ubiquitination and degradation. They start by showing that PB2 inhibits Type 1 IFN signaling in an over expression system and after viral infection. They go on to show that PB2 binds to JAK1 and promotes its degradation and K48 ubiquitination. They utilize various JAK1 mutants to demonstrate that K859 and K860 are the specific lysine residues that are targeted by PB2; they then investigate the effects of K859-860 mutation on adenoviral replication. They go on to show

that JAK1 interacts with PB2 and that PB2 from virulent influenza strains more strongly bind and repress JAK1, using PB2 constructs with mutations of 283M/526R. Finally, they use live virus with PB2 virulence-affecting mutations to show that mutation of PB2 to more/less virulent strains affects JAK1 expression, viral titers, and survival in vivo.

Overall, this study is novel and of interest to the community. Influenza polymerase has been linked to virulence through impaired type 1 interferon signaling, but the mechanisms are not well defined. The study incorporates extensive well-designed biochemical assays. I have several major concerns, however, that should be addressed by the authors prior to acceptance.

My most significant concern involves the presentation of pooled/collected quantified data for many of the experiments. Specifically, this paper involves a large number of Western blots and biochemical assays. While quantification numbers are given, there is no information about the number of replicates or statistical significance of each experiment. A graphical representation of the pooled data (rather than numbers on top of the bands) should be shown with the relevant statistics. The quantification data should also be clearly labeled -- specifically what normalization the investigators are using for each calculation. Some of the normalizations seem a bit confusing, i.e. in figure 2 the authors normalize to NP rather than actin but the legend does not clearly give the information about what this abbreviation stands for.

Figure 1g: panels a (mock) and b (Sev) appear to be switched. 1h-i: please normalize the phospho-STAT to total STAT

Figure 4: when the authors compare their K859R/K860R double mutant to other JAK1 mutants, in the PB2-CZ-transfected cells the positive controls don't seem to work. There is no clear difference in IB:His (JAK1) expression between PB2+/- samples for any of the conditions, including WT.

Figure 5: The authors state that "...replication of PR8 or CZ virus was inhibited in shJAK1 cells reconstituted with WT but not degradation-resistant JAK1... [indicating] that JAK1 ubiquitination/degradation at K859/860 residues is critical for efficient IAV replication..." In that case, a degradation-resistant JAK1 should prevent replication but in F5h-i, the cells transfected with shRNA-resistant/degradation-resistant JAK1 had no difference in viral replication compared with the untransfected cells. It also seems that they were compared with shRNA-resistant/WT JAK1 but I saw no comparison of WT vs. mutant JAK1 (not shRNA resistant) overexpression on viral replication. This is a major concern and should be addressed further

Figure 8: the authors state that their results demonstrate that PB2-mediated JAK1 degradation is critical for virus replication and transcription. However, they don't make the mechanistic link to JAK1 degradation. This is puzzling because they have degradation-resistant JAK1 mutants that they have created by mutating the relevant lysine residues. To definitively mechanistically implicate JAK1 as the driver of ISG induction, they should use these mutants with their various PB2 constructs.

Figure 10: Statistics are not provided for the in vivo survival curves in a-b. Histology scores are not calculated. 3 mice per group seems too exploratory; these experiments should be repeated with statistics provided.

Our point-by-point responses were shown as following.

Reviewer #1 (Remarks to the Author):

In "Influenza A virus Polymerase Protein PB2 Evades Antiviral Immunity by Inhibiting JAK1/STAT Signaling", Yang and colleagues show that PB2 can antagonize interferon signaling by mediating JAK1 degradation via the proteasome. The researchers further show that this function of PB2 increases influenza virus replication in vitro and in vivo. They identify the regions of JAK1 and PB2 that interact with each other and also identify the lysines in JAK1 that are ubiquitinated. A particularly interesting finding in this manuscript was that PB2s of H5 avian influenza viruses with residues 283I and 526K could degrade chicken JAK1 but not mammalian JAK1 and that this negatively impacted the replication of these viruses in mice and mammalian cells.

Overall, this manuscript is thorough and presents a lot of new findings. The experiments are well controlled, and the story is clear.

My only requirement is that the abstract be modified to better represent the findings of the paper. The authors should mention the mammalian versus avian adaptation results. There are also some spelling and grammatical errors that need to be corrected.

Response: Thanks for the comments and suggestions. We have revised the abstract as follows "Notably, the H5 subtype of highly pathogenic avian influenza virus with I283M/K526R mutations on PB2 increased the ability to degrade mammalian JAK1 and exhibited higher replicate efficiency in mammalian (but not avian) cells and mouse lung tissues, and caused greater mortality in infected mice" in lines 33-36. We have carefully checked the revised manuscript text and figures for spelling and grammatical

errors.

Reviewer #2 (Remarks to the Author):

In the current study, the authors investigate the interactions between influenza virus PB2 and the host cell JAK1 proteins. They demonstrate that PB2 inhibits the innate immune response by binding to and ubiquitinating JAK1. They explore the mechanism of this process and the differences between avian and mammalian viruses.

Major comments:

1. Having demonstrated the impact on innate responses in vitro, there is no in vivo data on this effect – the disease is more severe, but is IFN down or ISG in the lungs, could be measured by RT-PCR.

Response: Thanks for the suggestions. We repeated the in vivo study and determined the ISGs expression in the lungs. We added the results "On 2 dpi, the levels of *ISG15* (Fig. 10k) and *IFIT1* (Fig. 10l) mRNA in the mice infected with the rJY or rCZM were higher than the levels in the mice infected with the rJYM or rCZ." in lines 396-398.

2. How many times were the in vitro studies repeated?

Response: Thanks for the comment. Data represent one of three independent experiments or collected from at least three. We have added the description in the figure legends.

3. Figure 11 is not a clear demonstration of the mechanism of action. It has more dedicated to how JAK/Stat works than how PB2 inhibits it. A reworking of this figure, showing how the PB2 ubiquitinates the JAK (through what domain) and then how it

tags it for proteasome.

Response: Thanks for the suggestion. We have redrawn the working model in the revised manuscript (Fig. 11). IAV PB2 protein blocks JAK1-STAT signaling by targeting JAK1. IAV PB2 protein specifically mediates K48-linked ubiquitination and proteasomal degradation of JAK1, thereby suppressing JAK1-mediated signal transduction

4. The language throughout could be more clear, for example describing the degradation resistant proteins (line 247-249) is very unclear and needed multiple re-reads to understand.

Response: Thanks for the suggestion. We have re-worded the relevant sentence as "overexpression of wild-type JAK1 but not the K859/860R mutants JAK1 reduced virus replication in shJAK1 cells, suggesting that mutants of K859/860R damaged JAK1-mediated antiviral activity." in lines 224-226. We have carefully checked the revised manuscript text and figures for easy understanding.

Minor comments:

1. The figure legends are long and difficult to read because they include conclusions for each subpanel within them. Please make them more concise.

Response: Thanks for the suggestion. We have revised all figure legends to improve their conciseness.

2. In figure 3, what are up and down panels – do you mean upper and lower?

Response: Thanks for the suggestion. We revised as upper and lower in the legend of Figure 1, 4, 5, 7, 8 and Supplementary Figure 2, 6.

3. What does WCL stand for in the figures, please include an explanation

Response: Thanks for the suggestion. We added "WCL, Whole-cell lysates" in the legend of Figure 3.

4. Why are the UBQ plots shown at high and low exposure?

Response: Thanks for the comments. UBQ blots, especially in whole-cell lysates, are susceptible to being overexposed. Therefore, we included lower-exposure images for the blots. As suggested, we only kept one of two blots.

5. In figure 4C, can you put a hyphen in the K859R/K860R double mutant to indicate it is a double mutant. Why was the K859R or K860R single mutants not tested in the study in panel C.

Response: Thanks for the suggestion. We revised the double mutant as "K859/860R" in Fig. 4c. After ubiquitination modification online prediction of JAK1 by CPLM 1.0, we first carried out a double mutant of JAK1 in Fig. 4c, indicating that K859/860R double mutants abolished PB2-mediated degradation of JAK1. To further determine which lysine residue of JAK1 degradation by PB2, therefore, the K859R or K860R single mutant was tested in Fig. 4e. As shown in Fig. 4c to e, only simultaneous mutations of Lysine 859 and Lysine 860 to Arginine abolished PB2-mediated degradation of JAK1 (Fig. 4c-e).

6. Are the K48, K63 mutations in ubiquitin, it is not clear.

Response: Thanks for the suggestion. K48 or K63 mutants meant "K at indicated residue, and K at other residues was simultaneously mutated to arginines". We added the sentence in the legend of Figure 3.

7. Be consistent on SeV in the legend and the figure (in the figure it is put sev)

Response: Thanks. We have corrected it as SeV in Fig. 1.

Reviewer #3 (Remarks to the Author):

In this very thorough biochemical study, Yang et al describe a novel mechanism by which Influenza A polymerase protein PB2 suppresses antiviral responses by directly targeting JAK1 to promote its ubiquitination and degradation. They start by showing that PB2 inhibits Type 1 IFN signaling in an over expression system and after viral infection. They go on to show that PB2 binds to JAK1 and promotes its degradation and K48 ubiquitination. They utilize various JAK1 mutants to demonstrate that K859 and K860 are the specific lysine residues that are targeted by PB2; they then investigate the effects of K859-860 mutation on adenoviral replication. They go on to show that JAK1 interacts with PB2 and that PB2 from virulent influenza strains more strongly bind and repress JAK1, using PB2 constructs with mutations of 283M/526R. Finally, they use live virus with PB2 virulence-affecting mutations to show that mutation of PB2 to more/less virulent strains affects JAK1 expression, viral titers, and survival in vivo.

Overall, this study is novel and of interest to the community. Influenza polymerase has been linked to virulence through impaired type 1 interferon signaling, but the mechanisms are not well defined. The study incorporates extensive well-designed biochemical assays. I have several major concerns, however, that should be addressed by the authors prior to acceptance.

My most significant concern involves the presentation of pooled/collected quantified data for many of the experiments. Specifically, this paper involves a large number of Western blots and biochemical assays. While quantification numbers are given, there is no information about the number of replicates or statistical significance of each experiment. A graphical representation of the pooled data (rather than numbers on top of the bands) should be shown with the relevant statistics. The quantification data should also be clearly labeled -- specifically what normalization the investigators are using for each calculation. Some of the normalizations seem a bit confusing, i.e. in figure 2 the authors normalize to NP rather than actin, but the legend does not clearly give the information about what this abbreviation stands for.

Response: Thanks for the comments and suggestions. As suggested, a graphical representation of the pooled data has been shown with the relevant statistics and labeled the quantification data. Moreover, we have updated all of our graphs to show the individual data points and included all of the raw data in the required 'Source data file'.

Figure 1g: panels a (mock) and b (SeV) appear to be switched. 1h-i: please normalize the phospho-STAT to total STAT

Response: Thanks for the comments and suggestions. We double-checked Fig. 1g and found that panels a (mock) and b (SeV) were on the right panel. Since treatment of the supernatants of 293T cells [a (mock)] did not affect GFP expression, while treatment of the supernatants of 293T cells infected with SeV [b (SeV)] resulted in the induction of an IFN-mediated antiviral function that prevents replication of IFN-sensitive VSV-GFP, viral GFP expression was greatly reduced. As a suggestion, we have normalized the

phospho-STAT to total STAT in Fig. 1h, i.

Figure 4: when the authors compare their K859R/K860R double mutant to other JAK1 mutants, in the PB2-CZ-transfected cells the positive controls don't seem to work. There is no clear difference in IB:His (JAK1) expression between PB2+/- samples for any of the conditions, including WT.

Response: Thanks for the comment. As suggested, we repeated the experiments, provided new images, and normalized the JAK1 to Actin in Fig. 4c, d.

Figure 5: The authors state that "...replication of PR8 or CZ virus was inhibited in shJAK1 cells reconstituted with WT but not degradation-resistant JAK1... [indicating] that JAK1 ubiquitination/degradation at K859/860 residues is critical for efficient IAV replication..." In that case, a degradation-resistant JAK1 should prevent replication but in F5h-i, the cells transfected with shRNA-resistant/degradation-resistant JAK1 had no difference in viral replication compared with the untransfected cells. It also seems that they were compared with shRNA-resistant/WT JAK1 but I saw no comparison of WT vs. mutant JAK1 (not shRNA resistant) overexpression on viral replication. This is a major concern and should be addressed further

Response: Thanks for the comments and suggestions. As suggested, we compared the effects of JAK1 (WT) and JAK1 (K859/860R) on virus replication in shJAK1 cells (to eliminate the effect of endogenous JAK1 on viral replication). The results showed that the antiviral function of Δ JAK1 (K859/860R) was weakened relative to the Δ JAK1 (WT) when overexpressed in shJAK1 cells (Fig. 5h, i). To further address this major concern, we investigated the effect of JAK1 on the IFN-induced expression of ISGs.

The results revealed that IFN β -induced levels of *ISG15*, *IFIT1*, and *TAP1* mRNA were enhanced only in the shJAK1 cells transfected with JAK1 (WT)-expressing plasmid, not in those transfected with JAK1 (K859/860R)-expressing plasmid (Fig. 8g). The mutation of K859/860R of JAK1 may decrease the JAK1-mediated antiviral immune response, which could explain why reconstitution of the JAK1 K859/860R mutant did not restore the antiviral activity of JAK1 in the shJAK1 cells. These data suggest that K859/860 is the essential site for both JAK1 ubiquitination and JAK1 mediated ISGs antiviral activity. We added the explanation in discussion in lines 467-473.

Figure 8: the authors state that their results demonstrate that PB2-mediated JAK1 degradation is critical for virus replication and transcription. However, they don't make the mechanistic link to JAK1 degradation. This is puzzling because they have degradation-resistant JAK1 mutants that they have created by mutating the relevant lysine residues. To definitively mechanistically implicate JAK1 as the driver of ISG induction, they should use these mutants with their various PB2 constructs.

Response: Thanks for the comments and suggestions. As suggested, we first examined the effect of various mutated PB2 on JAK1(K859/860R)-mediated ISGs in the shJAK1 cells. The result showed that different PB2 mutations did not significantly affect JAK1(K859/860R)-mediated ISGs. Then, we checked the ISGs expression mediated by JAK1 (K859/860R) and found that JAK1 (K859/860R)-mediated ISGs levels were significantly reduced compared to JAK1 (WT) (Fig. 8g), which suggest that K859/860 is the essential site for both JAK1 ubiquitination and JAK1 mediated ISGs antiviral activity.

Figure 10: Statistics are not provided for the in vivo survival curves in a-b. Histology scores are not calculated. 3 mice per group seems too exploratory; these experiments should be repeated with statistics provided.

Response: Thanks for the comment. We provided statistics for the in vivo survival curves in Fig. 10a-b. Moreover, the relevant experiments in vitro were repeated. The lung tissue was collected from 2 and 5 dpi mice, with 5 mice in each group. At the same time, viral titers, HE staining, histochemical staining, and immunoblots were performed, and histology scores were calculated (Fig. 10). These results were similar to the previous study.

REVIEWERS' COMMENTS

Reviewer #2 (Remarks to the Author):

Thanks for addressing the comments

Reviewer #3 (Remarks to the Author):

The authors have sufficiently addressed all of my concerns, either by adding additional experimental data or by adjusting the text to more clearly explain the results as suggested by the other reviewers.

Our point-by-point responses were shown as following.

Reviewer #2 (Remarks to the Author):

Thanks for addressing the comments

Response: We thank for the comment.

Reviewer #3 (Remarks to the Author):

The authors have sufficiently addressed all of my concerns, either by adding additional experimental data or by adjusting the text to more clearly explain the results as suggested by the other reviewers.

Response: We would like to express our great appreciation to the reviewer for the comments.